# Flight trajectory prediction enabled by time-frequency wavelet transform

Zheng Zhang [1], Dongyue Guo [2], Shizhong Zhou[2], Jianwei Zhang[1,2] & Yi Lin [1,2] ✉

Accurate flight trajectory prediction is a crucial and challenging task in air traffic control, especially for maneuver operations. Modern data-driven methods are typically formulated as a time series forecasting task and fail to retain high accuracy. Meantime, as the primary modeling method for time series forecasting, frequency-domain analysis is underutilized in the flight trajectory prediction task. In this work, an innovative wavelet transform-based framework is proposed to perform time-frequency analysis of flight patterns to support trajectory forecasting. An encoder-decoder neural architecture is developed to estimate wavelet components, focusing on the effective modeling of global flight trends and local motion details. A real-world dataset is constructed to validate the proposed approach, and the experimental results demonstrate that the proposed framework exhibits higher accuracy than other comparative baselines, obtaining improved prediction performance in terms of four measurements, especially in the climb and descent phase with maneuver control. Most importantly, the time-frequency analysis is confirmed to be effective to achieve the flight trajectory prediction task.

With the continual development of the global economy, the air transportation demand has significantly increased across various industries, leading to a surge in flight traffic and airspace complexity. To optimize flight scheduling and improve operational efficiency, the traffic prediction is extensively studied to support air traffic management (ATM), including flight delay prediction[1,2], fuel consumption prediction[3,4], and flight trajectory prediction (FTP)[5,6]. Thanks to the supportive ability to the future trajectory-based operation (TBO), the FTP task is attracting increasing research attention for both the academic and industrial fields all over the world, including the Single European Sky ATM Research (SESAR)[7] and the Next Generation Air Transportation System (NextGen)[8]. The core idea of the TBO is to share future flight trajectories among traffic participants, enabling enhanced air-ground interconnection for a safe and effective air traffic control (ATC)[9]. An accurate prediction of the four-dimensional (4D) trajectory of aircraft serves as a fundamental technique to improve the predictability of air traffic for the TBO[10] to achieve downstream tasks, such as estimation of arrival time[11,12], conflict detection[13,14] and air traffic flow prediction[15–17].

The primary goal of the FTP task is to forecast the motion attributes that describe discrete trajectory points of an aircraft, such as longitude, latitude, altitude, speed, etc. Typically, the FTP task is defined as a multivariable time sequence forecasting problem considering current aircraft states and other operational and environmental factors. In terms of prediction horizons, the FTP task can be classified into short-term and long-term prediction tasks[18]. Short-term prediction task aims to provide accurate positional estimation to infer immediate traffic situation, primarily by modeling historical flight trajectories to predict future motion states. As for long-term prediction, additional external factors are required to support airspace operation planning and assessment, including flight intentions, meteorological conditions, wind speed, etc.

In this paper, we focus on short-term FTP task within a few minutes considering current flight trajectory. Existing approaches can be categorized into kinetics-and-aerodynamics, state-estimation, machine-learning, and deep-learning models[10]. Kinetics-and-aerodynamics models employ physical rules and handcrafted

[1]College of Computer Science, Sichuan University, Chengdu 610065 Sichuan, China. [2]National Key Laboratory of Fundamental Science on Synthetic Vision, Sichuan University, Chengdu 610065 Sichuan, China. ✉e-mail: yilin@scu.edu.cn

mathematical modeling to analyze the motion status of aircraft[19–23]. For the state-estimation models, the flight operation is modeled as a state transition process using state space theory[24–28]. For adapting to diverse flight patterns, the machine-learning models are able to learn hidden motion features from massive trajectory sequences[29–31]. However, kinetics-and-aerodynamics and state-estimation models suffer from insufficient generalization performance and impacts of environmental uncertainties, resulting in poor prediction accuracy. The prediction performance is also limited for the machine-learning models when a more complex maneuver control encountered. Thanks to the successful applications in natural language processing (NLP)[32], computer vision (CV)[33], automatic speech recognition (ASR)[34], and time series forecasting (TSF)[35] domains, the deep-learning models are also incorporated into the ATM research works by utilizing ATC operation data[36–39]. Currently, with the powerful data-fitting capabilities of neural networks, deep-learning models are regarded as promising tools to achieve the FTP task[40–42].

As mentioned above, the short-term FTP task is essentially a time series forecasting problem, implemented by modeling the complex and non-linear transition patterns of flight trajectory (several inter-related dynamic attributes at each time instant). Although the temporal modeling has been applied to capture the autoregressive properties of dynamic attributes[43–46], it is still a challenging task to examine the underlying flight patterns in sufficient details. In general, the primary dynamics results from the aircraft maneuvering during the climb and descent phase, as well as the intention-driven operations. Considering aircraft safety and passenger comfort, the intensity of maneuver controls is restricted to obtain limited saliency on the temporal trajectory sequence, resulting in the inability to capture such maneuvering patterns for conventional models. A practical resolution is to leverage frequency-domain information, which enables capturing informative patterns from time-frequency features to support the FTP task. In the TSF field, the frequency-domain analysis is applied to break time series down to promote in-depth inference[47–49]. Considering the time series nature of the FTP task, the frequency information is also involved in the trajectory attributes. For instance, the longitude and latitude are always changing in a general evolution direction from the origin to the destination during the flight operation, so the longitude and latitude components of the trajectory can be considered to illustrate global flight trends. Driven by flight intention (turn, climb,

descent, etc.), the time series of longitude, latitude and altitude will react to the corresponding changes, as the local details of the aircraft motion. However, the current method primarily focused on modeling in the time domain, and time-frequency analysis is still a virgin task in the FTP research, without the delicate decomposition on frequency details. Therefore, inspired by successful applications in other TSF tasks, it is believed that time-frequency analysis is a promising solution to capture the underlying patterns of flight trajectories, allowing us to achieve the FTP task from a more delicate perspective.

To this end, an innovative framework, i.e., Wavelet Transform-based Flight Trajectory Prediction (WTFTP), is proposed to perform wavelet analysis[50] to model global flight trends and local aircraft motion details. The architecture of the proposed framework is illustrated in Fig. 1. The wavelet analysis is able to decompose the input flight trajectory into the wavelet coefficients at different time and frequency resolutions using discrete wavelet transform (DWT). Inversely, these wavelet coefficients can also be applied to reconstruct the raw trajectory via an inverse discrete wavelet transform (IDWT) module. For the FTP task, the fundamental requirement is to generate the optimal wavelet coefficients from the input flight trajectory, which has the ability to support the IDWT reconstruction to predict the next trajectory point.

To implement this, an encoder-decoder neural architecture is proposed to learn latent temporal features from the input trajectory sequence and project these features into the wavelet domain, i.e., generating wavelet coefficients of both historical and future trajectories by using different scale-oriented decoders. The estimated coefficients are further passed through an IDWT module to achieve the FTP task. In this context, wavelet components refer to a set of wavelet coefficients obtained from decomposed motion attributes. For each motion attribute, the wavelet components are inferred from different decomposition paths and located at different scales. In general, high-frequency components represent local details of the flight trajectory, while low-frequency ones indicate global trends. To support the wavelet reconstruction procedure, a wavelet attention module is innovatively designed to capture discriminative transition patterns by learning scale-oriented coefficients from input trajectory sequences. In the training stage, the actual wavelet coefficients are utilized as supervised information to update neural parameters. The experimental results demonstrate the proposed WTFTP framework achieve

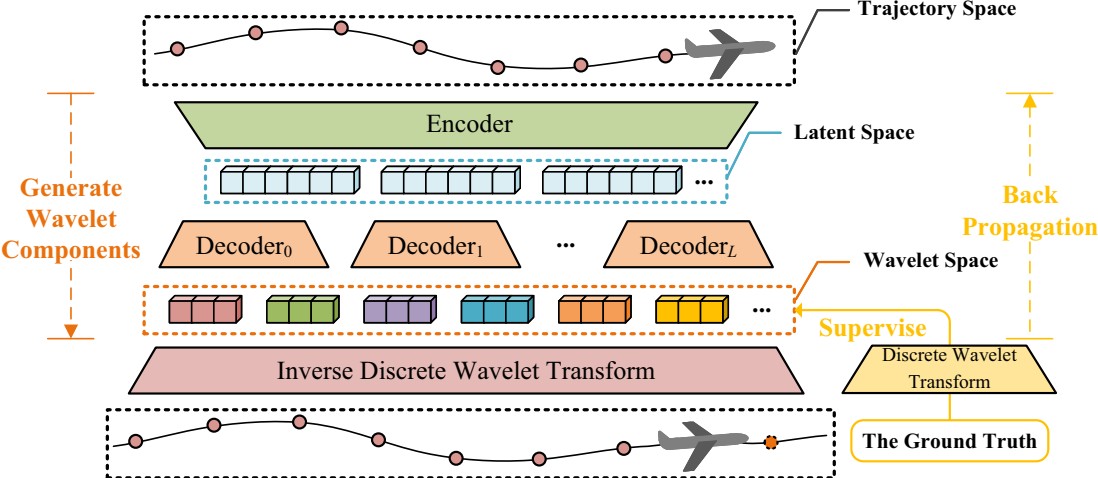

**Fig. 1 | The overall architecture of the proposed wavelet transform-based flight trajectory prediction framework.** The orange dash line denotes the prediction objective of the proposed framework, i.e., wavelet components, which are applied to perform the inverse discrete wavelet transform for reconstructing historical sequence and predicting the next state of the aircraft. From upper to lower, the historical trajectory sequence is transformed into latent temporal features that are fed into multiple decoders to extract scale-oriented features for generating wavelet components. The yellow solid line indicates supervision information, i.e., the real wavelet components (obtained by discrete wavelet transform of the real historical trajectory sequence and the next-instant trajectory point) serve as supervised information.

less than 400-meter three-dimensional deviation error and robust prediction performance in different flight phases (i.e., cruise, climb and descent). By incorporating the wavelet analysis into the FTP task, a time-frequency perspective is provided to perform the pattern recognition for different time and frequency scales, making it highly effective for multi-resolution analysis to enhance the FTP performance. In addition, the proposed approach also enables us to extract underlying dynamic properties from these multi-scale components, providing a more comprehensive representation of flight trajectories over the traditional time-domain representation.

In this work, the proposed framework contributes the flight trajectory prediction task in the following ways:

(1) A wavelet-based time-frequency framework is innovatively proposed to achieve the FTP task. Compared to previous works, the proposed framework has greatly improved flight trajectory prediction by incorporating time-frequency analysis to capture dynamic characteristics of trajectories.

(2) An encoder-decoder deep-learning architecture is proposed to generate wavelet coefficients, in which global flight trends and local motion details at different scales are separately modeled to support the IDWT procedure for the FTP task.

(3) A wavelet attention module is designed in each decoder to exploit scale-oriented underlying patterns from historical trajectory sequences and enhance the learning ability towards flight patterns at different scales to promote the prediction performance.

(4) The proposed approach is validated on real-world data and the experimental results demonstrate the performance advantages over other competitive baselines, especially in the climb and descent phases with maneuver control. All the proposed technical modules contribute to desired performance improvements. The results also confirm the effectiveness of the time-frequency analysis for the FTP task.

## Results

### Task overview

The FTP task is generally defined as a TSF problem. Given the attribute vectors of the past $M$ trajectory points $\{\mathbf{P}_i \in \mathbb{R}^d | i = N-M, N-M+1, \cdots, N-1\}$, the primary objective is to predict the attribute vectors of the future trajectory point $\mathbf{P}_N$. Here we define $\mathbf{P}_i$ as follows:

$$\mathbf{P}_i = [Lon_i, Lat_i, Alt_i, Vx_i, Vy_i, Vz_i]^\mathsf{T} \tag{1}$$

where $Lon$, $Lat$, $Alt$, $Vx$, $Vy$, $Vz$ correspond to the longitude, latitude, altitude and velocities along the previous three dimensions, respectively. The superscript T denotes matrix transposition. A non-linear function $f(\cdot)$ is expected to be learned and estimate the next status $\hat{\mathbf{P}}_N$:

$$\hat{\mathbf{P}}_N = f(\mathbf{P}_{N-M:N-1}) \tag{2}$$

$$\mathbf{P}_{N-M:N-1} = [\mathbf{P}_{N-M}, \mathbf{P}_{N-M+1}, \cdots, \mathbf{P}_{N-1}] \tag{3}$$

The proposed WTFTP framework implements the trajectory prediction by performing the IDWT procedure to reconstruct the historical trajectory sequence and predict the trajectory point for the next instant, in which the wavelet components are generated by a neural architecture to consider different frequency scales. Specifically, the low-frequency components can be considered as global flight trends, which imply the general orientation of the trajectory, while the high-frequency components represent the local motion details at different scales to capture maneuvering patterns of the aircraft. By applying these wavelet components to supervise the learning procedure, the WTFTP is able to identify different flight patterns from trajectory sequence to support the reconstruction and prediction procedure.

Mathematically, the prediction process of the WTFTP can be described as:

$$[\hat{\mathbf{P}}_{N-M:N-1}, \hat{\mathbf{P}}_N] = \text{WTFTP}(\mathbf{P}_{N-M:N-1}^\mathsf{T}) \tag{4}$$

More details about the inference procedure of the proposed WTFTP framework can be found in Supplementary Section 2.2.

### Dataset and preprocessing

In this work, the raw flight trajectories are collected by multi-source Secondary Surveillance Radar (SSR) and Automatic Dependent Surveillance-Broadcast (ADS-B) from a real-world ATC system in China. The flight trajectory dataset is formulated by fusing the multi-source flight trajectories to validate the proposed approach. In addition, the preprocessing steps are conducted to enhance the data quality, including data check, multi-source data parsing, track point and flight registering, multi-source data alignment, data filtering, and fusion. The trajectory dataset covers about 45 days. The timestamp, position and speed attributes in the 3D earth space are parsed from raw binary data to build our experimental dataset.

The update interval of the trajectory in this dataset is 20 s. The trajectory attributes are normalized using max-min normalization to unify data scales and ranges (longitude and latitude: degree, altitude: 10 m, speed: kilometers per hour). To evaluate the model performance, the trajectories in the first 40 days are selected as the training dataset, and the following one day is dedicated to fine-tuning hyperparameters, and the trajectories of the last four days are formulated as a test set. More detailed descriptions of the dataset are provided in Supplementary Section 1.1.

### Evaluation metrics

In this work, a total of four measurements are considered to evaluate the performance of prediction models, including root of mean squared error (RMSE), mean absolute error (MAE), mean relative error (MRE) and mean deviation error (MDE). The commonly used metrics in the FTP task are RMSE, MAE, and MRE, as shown below:

$$RMSE_i = \sqrt{\frac{1}{N}\sum_{j=1}^{N}(\mathbf{P}_{i,j} - \hat{\mathbf{P}}_{i,j})^2} \tag{5}$$

$$MAE_i = \frac{1}{N}\sum_{j=1}^{N}|\mathbf{P}_{i,j} - \hat{\mathbf{P}}_{i,j}| \tag{6}$$

$$MRE_i = 100\% \times \frac{1}{N}\sum_{j=1}^{N}\left|\frac{\mathbf{P}_{i,j} - \hat{\mathbf{P}}_{i,j}}{\mathbf{P}_{i,j}}\right| \tag{7}$$

where $N$ represents the total number of the test set. $\mathbf{P}_{i,j}$ is the $i$-th attribute of the real trajectory point for the $j$-th sample and $\hat{\mathbf{P}}_{i,j}$ is the corresponding predicted value.

As the RMSE, MAE and MRE are useful metrics for assessing prediction performance on a single motion attribute, in this work, the 3D deviation distance between the predicted and ground-truth trajectory point is also measured to consider overall model performance. To resolve the differences in measurements and statistical spans between longitude, latitude, and altitude in the WGS-84 coordinate system, the MDE metric, based on the Euclidean distance between predicted and actual points in earth-centered and earth-fixed (ECEF) coordinate system, is proposed to measure the overall prediction performance of

the FTP model as shown below:

$$MDE = \frac{1}{N}\sum_{j=1}^{N}\sqrt{\sum_{i}^{3}(pos_{i,j} - \hat{pos}_{i,j})^2} \qquad (8)$$

$$\begin{cases} pos_{1,j} = (PR(lon_j) + alt_j)\cos(lon_j)\cos(lat_j) \\ pos_{2,j} = (PR(lon_j) + alt_j)\cos(lon_j)\sin(lat_j) \\ pos_{3,j} = \left(\frac{b^2}{a^2}PR(lon_j) + alt_j\right)\sin(lon_j) \\ PR(\cdot) = \dfrac{a}{\sqrt{1 - \left(1 - \frac{b^2}{a^2}\right)\sin^2(\cdot)}} \end{cases} \qquad (9)$$

where $pos_{1,j}$, $pos_{2,j}$ and $pos_{3,j}$ are converted coordinates of the actual position components (i.e., longitude, latitude and altitude) $lon_j$, $lat_j$ and $alt_j$, respectively. $\hat{pos}_{1,j}$, $\hat{pos}_{2,j}$ and $\hat{pos}_{3,j}$ are converted coordinates of the predicted position components $\hat{lon}_j$, $\hat{lat}_j$ and $\hat{alt}_j$ by using the same conversion equations as shown in Eq. (9), respectively. $a = 6378.137$ kilometers (km) and $b = 6356.752$ km are the equatorial radius and the polar radius of the earth, respectively. PR(·) is the distance from the surface to the line between the north and south poles along the ellipsoid normal. The predicted and target positions are converted into the ECEF coordinate system with the same measurement, kilometers, by using Eq. (9). Finally, the MDE can be computed by their Euclidean distance. Considering that timestamp, longitude, latitude and altitude serve as crucial attributes in general 4D-FTP tasks, the prediction performance is primarily investigated on the position components due to the fixed time interval in this work. The speed is considered as the auxiliary attributes to learn flight patterns in the WTFTP framework.

## Model configurations

All experiments are built and conducted on PyTorch 1.4[51]. The PyTorch implementation of wavelet transform[52] is utilized to support the procedure of the DWT and IDWT procedures. The Adam optimizer is selected to update trainable parameters with a learning rate initialized as 0.001 and decayed by a rate of 0.5 every 10 epochs. More details about experiment settings can be found in Supplementary Section 1.2. The model configurations are optimized by training on the small data set and fine-tuned by the validation set. More details of configurations can be found in Supplementary Table 1.

## Baselines

To validate the proposed WTFTP framework in the FTP task, several baseline models with different model architectures and technical frameworks are selected to compare the model performance on the test dataset, as shown below:

A1 Vanilla LSTM: This is an RNN-based predictor proposed in[43] with the LSTM networks modeling trajectory points. The input and output embedding layer with fully connected networks are employed to implement feature projections.

A2 TCN: This is a sequential modeling architecture proposed in[53], and the causal convolution mechanism can more effectively model temporal information. The TCN applied in the FTP is recently studied[54] and we also validate its performance in our data set.

A3 CNN LSTM: Based on the vanilla LSTM, the CNNs are applied to extract the spatial information and further combined with the LSTM networks to achieve the FTP task[44].

A4 Transformer: By referring to other works in the CV, TSF, ASR and NLP fields, the Transformer architecture[32] is also selected as the baseline to achieve the FTP task in a non-autoregressive manner.

A5 FlightBERT: This is a Transformer-based flight predictor proposed in[41], in which the binary encoding representation is proposed to enhance the feature extraction. This work achieves the FTP task as a multi-binary classification problem.

## Overall performance

The experimental results of the proposed approach and other selective baselines are reported in Table 1, in terms of the proposed four measurements. In general, the proposed WTFTP framework outperforms other baselines and achieves the best performance in all metrics of longitude, latitude and altitude (LLA) except the RMSE of altitude, which showcases its performance advantages and also confirms the effectiveness of time-frequency analysis in the FTP task. Thanks to the ability of time-frequency analysis and in-depth modeling of flight trends and motion details, the WTFTP framework achieves a relative reduction of at least 30% MAE and 20% MRE in the longitude and latitude dimensions compared to the best results of these baselines (FlightBERT). Furthermore, the practicality of the WTFTP framework is greatly enhanced since it reduces the MDE by approximately 35%, achieving less than 400-meter prediction error. The results in the MDE enable the proposed approach to be a promising solution in real-world applications, thanks to small deviations of the predicted trajectory points. From the experimental results, the following conclusions can also be drawn to understand the proposed approach:

(1) As a basic deep-learning model only considering the temporal modeling in the FTP task, A1 suffers from the largest prediction errors, i.e., 0.9472 km in the MDE metric, which makes it challenging to support delicate trajectory operation management. To improve the prediction accuracy, in A2, the causal convolution mechanism is applied to effectively establish long-range time-series relationships of historical trajectory, resulting in

**Table 1 | The overall performance evaluation**

| Models | MAE↓ | | | MRE(%)↓ | | | RMSE↓ | | | MDE↓ |
|---|---|---|---|---|---|---|---|---|---|---|
| | Lon | Lat | Alt | Lon | Lat | Alt | Lon | Lat | Alt | |
| A1 | 0.0056 | 0.0059 | 1.36 | 0.0030 | 0.0110 | 0.25 | 0.0163 | 0.0142 | 8.55 | 0.9472 |
| A2 | 0.0052 | 0.0054 | 1.32 | 0.0049 | 0.0194 | 0.27 | 0.0151 | 0.0138 | **8.13** | 0.8794 |
| A3 | 0.0051 | 0.0050 | 1.36 | 0.0048 | 0.0177 | 0.27 | 0.0164 | 0.0139 | 8.91 | 0.8299 |
| A4 | 0.0049 | 0.0047 | 1.21 | 0.0046 | 0.0169 | 0.24 | 0.0148 | 0.0128 | 8.17 | 0.8003 |
| A5 | 0.0039 | 0.0033 | 1.36 | 0.0029 | 0.0103 | 0.30 | 0.0558 | 0.0486 | 10.59 | 0.5910 |
| WTFTP | **0.0025** | **0.0022** | **1.14** | **0.0023** | **0.0078** | **0.23** | **0.0148** | **0.0125** | 8.91 | **0.3855** |

A1–A5 represent the baselines: Vanilla LSTM, TCN, CNN LSTM, Transformer and FlightBERT, respectively.
The Lon, Lat and Alt stand for longitude, latitude and altitude, respectively.
In the MAE and RMSE metrics, the Lon and Lat are measured in degrees, and the Alt is measured in 10 m. The MDE is measured in kilometers.
The bold ones denote the best performance on the corresponding metric.
↓ represents minimization indicators.

slightly better regression results compared to those of vanilla LSTM. Considering that only the temporal modeling in the FTP task fails to obtain desired accuracy, the results confirm that both spatial and temporal features are required for modeling the trajectory sequences to achieve a high-confidence FTP task.

(2) Considering the requirements of the spatial and temporal modeling, in A3, the convolution mechanism and recurrent inference are combined to achieve the FTP task, resulting in slight performance improvement due to the temporal-spatial modeling capability. However, this model also fails to capture intrinsic flight patterns at different scales and provide the desired prediction performance, without in-depth feature extraction towards global trends and local details of the trajectory sequence.

(3) To enhance the learning capability for flight patterns, the self-attention mechanism in A4 is designed to correlate historical trajectory points and extract semantic representations of the sequence, resulting in more desirable prediction results. Meanwhile, it is also noted that A4 is able to provide a robust prediction in terms of the RMSE metrics on all the LLA dimensions. The results can be attributed that the self-attention mechanism highlights the trajectory characteristic at significant historical steps, further enhancing the prediction accuracy.

(4) In order to explore effective high-dimensional trajectory features and further promote the overall capability of Transformer, the binary encoding representation and attribute correlation attention in A5 (FlightBERT) are proposed to achieve the FTP task, which provides significant performance improvements in the MAE and MRE metrics for all attributes (except altitude) over A1-A4 baselines, and over 26% in the MDE metric. As demonstrated in the original paper, the inferior performance on the altitude dimension is also caused by the high-bit prediction error of the binary encoding. Although sequential inference and feature characterization contribute considerable performance improvements, the Transformer-based models A4 and A5 suffer from limited prediction accuracy. This is due to the deficiency of time-frequency analysis, resulting in inadequate learning ability towards the underlying flight patterns.

Fortunately, in the proposed WTFTP framework, the time-frequency representations of wavelet components (WTCs) enable it to sufficiently examine the slow and fast dynamic properties of the flight trajectory and thus harvest the best performance over other models. To be specific, the improvement of the RMSE metric is relatively smaller compared to other models. It is primarily because the modeling of high-frequency components inevitably includes the estimation noise, and thus has an impact on prediction stability. Moreover, the RMSE of altitude is higher in the WTFTP than that in selected baselines. The performance reduction is also caused by the over-modeling towards fast dynamics on the altitude dimension, since the flight cruise is the primary phase in the civil aviation operation process with relatively little maneuver control over altitude (i.e., abundant slow dynamics on the altitude dimension). However, the prediction gap in altitude is still under 10 meters, and the MAE and MRE of altitude still outperform other models, which also validates the modeling of the ability of the proposed approach. In summary, the proposed WTFTP framework harvests the highest performance and confirms the effectiveness of time-frequency analysis in the FTP task, which also supports the motivation of this work.

To further evaluate the performance of the WTFTP framework and baselines, experiments in different flight phases (including cruise, climb and descent) are also investigated and the results are summarized in Table 2. In general, the proposed WTFTP framework has the ability to provide a robust prediction performance in all flight phases. From the results, the following conclusions can be drawn:

(1) In the cruise phase, it can be seen that the WTFTP outperforms baselines in most metrics, in which the altitude prediction is affected by the same problem as discussed above. Although the RMSE of longitude is also impacted by prediction noise (about a 0.001-degree gap from the best baseline), the WTFTP framework is still able to achieve a satisfactory overall performance in the MDE metric.

(2) In the climb and descent phases, compared to their metrics in the cruise phase, a common phenomenon can be observed that the comparative baselines suffer from severe performance gap in the comprehensive MDE metrics due to the situations of intention change and maneuver control. Thanks to the capability of time-frequency analysis and in-depth feature extraction towards global flight trends and local motion details, the proposed WTFTP approach is capable of effectively capturing intrinsic evolution patterns of trajectory to provide a robust prediction performance and can still guarantee prediction accuracy in such complex situations (0.3405 km of the MDE in the cruise phase v.s. 0.4846 km and 0.3753 km in the climb and descent phase).

As we know, the climb and descent in the terminal airspace is the performance bottleneck of the FTP task, which is also the primary focus of current FTP methods. This work contributes a high-confidence predictive stability to advance the FTP application into the real-world industrial level, which further confirms the effectiveness and clarifies the necessity of the proposed WTFTP and time-frequency analysis in the FTP task.

## Ablation study

To further study the performance contributors of the proposed approach, including different levels of wavelet analysis and the wavelet attention module, and verify the effectiveness of time-frequency analysis in the FTP task, the following configurations are considered as the ablation experiments, as shown below:

B1 2-level WTFTP: In this case, we study the prediction performance based on a higher level of wavelet analysis, and other hyperparameters are the same as the WTFTP.

B2 3-level WTFTP: Similar to B1, we set the level of wavelet analysis to 3 in this case.

C1 WTFTP without WAtt: In this case, we remove the WAtt module to study performance improvements of the WAtt module and the effectiveness of time-frequency analysis in the FTP task. This model relies on an autoregressive inference to generate WTCs.

C2 2-level WTFTP without WAtt: Similar to C1 with 2-level wavelet analysis.

C3 3-level WTFTP without WAtt: Similar to C1 with 3-level wavelet analysis.

The experimental results of the ablation studies are reported in Table 3 and the following conclusions can be drawn:

(1) For the levels of the wavelet analysis, we can see that the proposed approach has the ability to obtain comparable performance among the experiments, i.e., 2-level of wavelet analysis indeed improves the final FTP performance, but performance degradation will be encountered for the 3-level wavelet analysis. Specifically, compared to the WTFTP framework, B1 exhibits enhanced performance for most performance indicators due to its higher level of wavelet analysis. The higher level of wavelet analysis also results in the performance improvement of C2 over C1 by removing the WAtt modules. The B1 outperforms the WTFTP primarily because a higher level of wavelet analysis provides more detailed dynamic characteristics, i.e., more high-frequency WTCs, about the flight trajectory, which allows decoders to finely model the motion properties of the aircraft. However, the performance of B2 is inferior to the WTFTP framework, except the MAE and RMSE metrics on the altitude dimension. Similarly, C3 suffers from performance reduction caused by an excessively high level of wavelet analysis after

**Table 2 | Performance comparison in different flight phases**

| Phases | Models | MAE↓ | | | MRE(%)↓ | | | RMSE↓ | | | MDE↓ |
|---|---|---|---|---|---|---|---|---|---|---|---|
| | | Lon | Lat | Alt | Lon | Lat | Alt | Lon | Lat | Alt | |
| Cruise | A1 | 0.0043 | 0.0050 | 0.58 | 0.0041 | 0.0180 | 0.07 | 0.0166 | 0.0142 | 5.90 | 0.7789 |
| | A2 | 0.0043 | 0.0044 | 0.59 | 0.0040 | 0.0159 | 0.07 | 0.0155 | 0.0136 | 5.47 | 0.7173 |
| | A3 | 0.0041 | 0.0042 | 0.62 | 0.0038 | 0.0152 | 0.07 | 0.0169 | 0.0144 | 6.38 | 0.6837 |
| | A4 | 0.0045 | 0.0050 | 0.56 | 0.0042 | 0.0173 | 0.06 | **0.0155** | 0.0140 | 5.58 | 0.7954 |
| | A5 | 0.0042 | 0.0033 | **0.09** | 0.0041 | 0.0133 | **0.01** | 0.0612 | 0.0458 | **1.90** | 0.5479 |
| | WTFTP | **0.0022** | **0.0019** | 0.36 | **0.0020** | **0.0070** | 0.04 | 0.0164 | **0.0136** | 6.35 | **0.3405** |
| Climb | A1 | 0.0084 | 0.0079 | 3.16 | 0.0079 | 0.0277 | 0.57 | 0.0118 | 0.0111 | 4.42 | 1.3473 |
| | A2 | 0.0079 | 0.0087 | 3.12 | 0.0075 | 0.0306 | 0.57 | 0.0116 | 0.0118 | 4.46 | 1.3701 |
| | A3 | 0.0070 | 0.0067 | 3.12 | 0.0066 | 0.0234 | 0.57 | 0.0104 | 0.0092 | 4.30 | 1.1225 |
| | A4 | 0.0060 | 0.0070 | **3.00** | 0.0057 | 0.0245 | 0.56 | 0.0084 | 0.0094 | 4.18 | 1.0802 |
| | A5 | 0.0046 | 0.0040 | 7.46 | 0.0044 | 0.0144 | 1.45 | 0.0521 | 0.0512 | 28.08 | 0.9032 |
| | WTFTP | **0.0030** | **0.0028** | 3.01 | **0.0028** | **0.0098** | **0.55** | **0.0043** | **0.0038** | 4.16 | **0.4846** |
| Descent | A1 | 0.0066 | 0.0060 | 1.94 | 0.0062 | 0.0211 | 0.55 | 0.0097 | 0.0085 | 2.89 | 1.0363 |
| | A2 | 0.0061 | 0.0062 | 1.91 | 0.0057 | 0.0222 | 0.54 | 0.0086 | 0.0086 | 2.82 | 1.0128 |
| | A3 | 0.0060 | 0.0052 | 2.00 | 0.0056 | 0.0183 | 0.56 | 0.0088 | 0.0073 | 2.95 | 0.9196 |
| | A4 | 0.0050 | 0.0060 | 2.09 | 0.0047 | 0.0211 | 0.63 | 0.0075 | 0.0080 | 2.95 | 0.9175 |
| | A5 | 0.0041 | 0.0047 | 3.94 | 0.0040 | 0.0127 | 0.92 | 0.0504 | 0.0726 | 18.85 | 0.7996 |
| | WTFTP | **0.0024** | **0.0021** | **1.81** | **0.0023** | **0.0074** | **0.50** | **0.0040** | **0.0030** | **2.73** | **0.3753** |

A1–A5 represent the baselines: Vanilla LSTM, TCN, CNN LSTM, Transformer and FlightBERT, respectively.

The Lon, Lat and Alt stand for longitude, latitude and altitude, respectively.

In the MAE and RMSE metrics, the Lon and Lat are measured in degrees, and the Alt is measured in 10 m. The MDE is measured in kilometers.

The bold ones denote the best performance on the corresponding metric.

↓ represents minimization indicators.

**Table 3 | Experimental results of ablation studys**

| Models | WAtt | MAE↓ | | | MRE(%)↓ | | | RMSE↓ | | | MDE↓ |
|---|---|---|---|---|---|---|---|---|---|---|---|
| | | Lon | Lat | Alt | Lon | Lat | Alt | Lon | Lat | Alt | |
| WTFTP | w/ | 0.0025 | 0.0022 | 1.14 | 0.0023 | 0.0078 | **0.23** | 0.0148 | 0.0125 | 8.91 | 0.3855 |
| B1 | | **0.0024** | **0.0021** | 1.15 | **0.0022** | **0.0075** | 0.24 | **0.0134** | **0.0117** | 8.90 | **0.3727** |
| B2 | | 0.0032 | 0.0032 | **1.12** | 0.0030 | 0.0113 | 0.24 | 0.0140 | 0.0123 | **8.59** | 0.5271 |
| C1 | w/o | 0.0032 | 0.0029 | 1.17 | 0.0030 | 0.0105 | 0.24 | 0.0159 | 0.0131 | 8.93 | 0.5090 |
| C2 | | 0.0028 | 0.0028 | 1.26 | 0.0026 | 0.0099 | 0.26 | 0.0140 | 0.0122 | 9.07 | 0.4642 |
| C3 | | 0.0032 | 0.0031 | 1.25 | 0.0030 | 0.0110 | 0.25 | 0.0140 | 0.0119 | 8.95 | 0.5186 |

B1 and B2 represent 2- and 3-level WTFTP with wavelet attention module, respectively. C1–C3 represent 1-, 2-, and 3-level WTFTP without wavelet attention module.

The Lon, Lat and Alt stand for longitude, latitude and altitude, respectively.

In the MAE and RMSE metrics, the Lon and Lat are measured in degrees, and the Alt is measured in 10 m. The MDE is measured in kilometers.

Notaion w/ and w/o indicate the WAtt module is included in the decoder or not.

The bold ones denote the best performance on the corresponding metric.

↓ represents minimization indicators.

removing the WAtt modules. The primary reason for the performance degradation is the over-short length of WTCs in B2 and C3, which undermines the temporal modeling of the WTFTP framework. As demonstrated in Eqs. (19)–(21), the shortest length of WTCs is 2 in B2 and C3, thereby impacting the decoders to learn the evolution patterns of the flight trajectory.

(2) As to the beneficial effects of the WAtt module, all experimental results confirm the effectiveness of the proposed WAtt module in capturing scale-oriented features and enhancing the contextual representation of trajectory sequences at different scales to precisely predict WTCs. Specifically, in the cases of the same level, by incorporating the WAtt module into the proposed approach, lower prediction errors are obtained for all metrics. The performance reduction of 3-level wavelet analysis in B2 can also be attributed to deteriorated temporal modeling caused by the

over-short length of WTCs. Thanks to the capability of the WAtt module to improve particular correlations between historical trajectory points and future motion properties in certain corresponding scales, the WTFTP framework is able to provide required trajectory patterns and therefore outperforms C2 for all position components in terms of the proposed metrics.

(3) In addition, it can also be seen that, even without the WAtt module, the proposed wavelet framework (implemented by a simple encoder-decoder structure in C1–C3) also harvests better performance over other baselines, particularly in terms of latitude and longitude metrics as well as the MDE, as shown in Tables 1 and 3. The improvement primarily results from the capability of time-frequency analysis to capture diverse flight patterns of global trends and local details, whereby the WTFTP framework achieves the extraction of in-depth features related

to multi-resolution aircraft motion properties. As demonstrated by the MDE metric in Table 3, the time-frequency analysis-based models accurately predict flight trajectory with smaller deviations, which validates their robustness and practical performance and further confirms the effectiveness of time-frequency analysis in the FTP task.

To provide a more comprehensive understanding of interpretability for the WAtt module, attention scores for the WTFTP and B1 are visualized in Fig. 2a–c. Specifically, Fig. 2a, b display the attention scores of $WTC_0$ and $WTC_1$ in the decoders of the WTFTP framework, respectively, while Fig. 2c–e illustrate the attention scores of $WTC_0$, $WTC_1$, and $WTC_2$ in the decoders of B1. The detailed explanation of the subscripts of the WTC can be found in Section Time-frequency features of flight trajectory, which represents different scales. From Fig. 2a, c, attention scores for generating $WTC_0$ assign higher weights at the end of the historical trajectory, which confirms intuitive patterns of the flight trajectory, i.e., the trend of the next trajectory point highly correlates with the last historical trajectory. In addition, Fig. 2b reveals that $WTC_1$ learns higher attention to the last two historical trajectory points of the high-frequency component, in which two neighbor points can reflect local change details of the flight trajectory. Figure 2e

further demonstrates that $WTC_2$ can provide the highest frequency activations for more significant latter motion details of historical trajectory. However, due to the 2-level wavelet analysis in B1, two sets of attention scores for high-frequency patterns are entirely distinct from Fig. 2b. In particular, some earlier steps of the trajectory are assigned to prominent attention scores in Fig. 2d, e. Most importantly, although both $WTC_1$ and $WTC_2$ are to capture detailed local patterns of the flight trajectory, their distributions of attention scores are complementary with each other. Therefore, it is believed that additional abstract representations in the frequency domain examine fine-grained dynamic detail features at earlier time steps, ultimately extracting more in-depth flight patterns to enhance the prediction performance.

## Case study of complex airspace scene

To study the prediction performance in complex scenes, a representative flight path is selected to implement visualization. The specific flight journey is from the location with the longitude and latitude coordinates of around (104.15, 30.50) and to (102.19, 27.84), where a flying circle of the approach phase is caused due to traffic flow control near the arrival airport, as presented in Fig. 3a. The 3D visualization is also provided to support the evaluation in Fig. 3b and shows that the

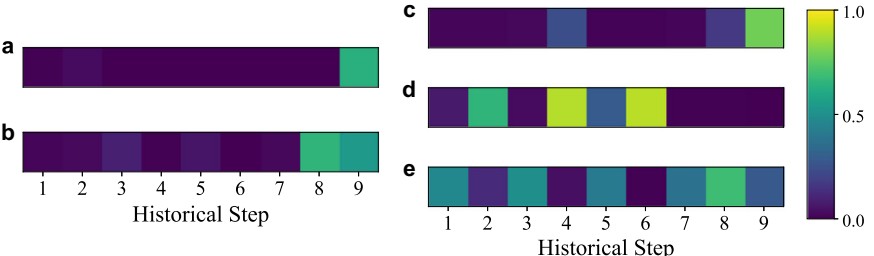

**Fig. 2 | Attention scores obtained in decoders of one-level and two-level WTFTP. a** Attention scores of one-level WTFTP for generating $WTC_0$. **b** Attention scores of one-level WTFTP for generating $WTC_1$. **c** Attention scores of two-level

WTFTP for generating $WTC_0$. **d** Attention scores of two-level WTFTP for generating $WTC_1$. **e** Attention scores of two-level WTFTP for generating $WTC_2$. All score values are provided as a source data file.

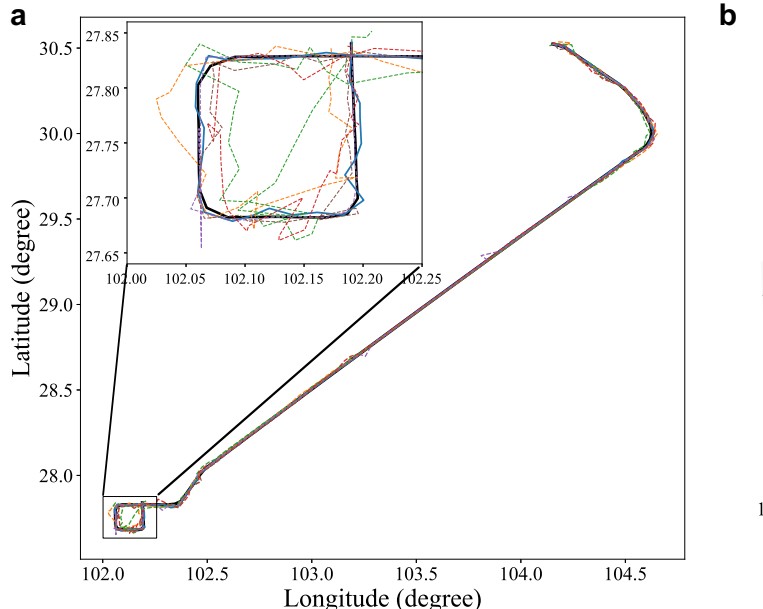

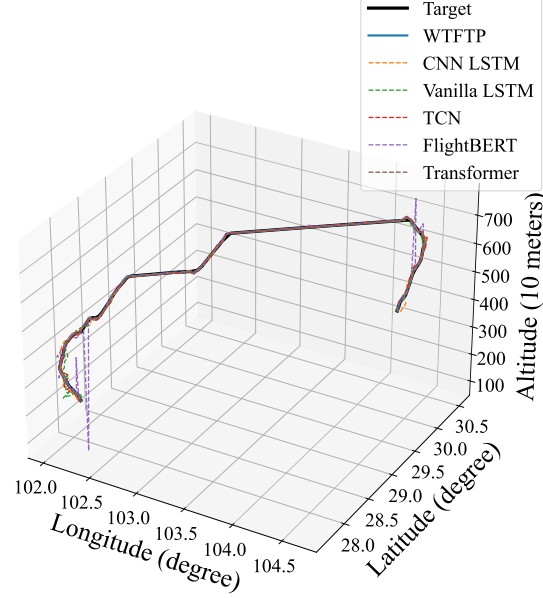

**Fig. 3 | Visualization of the selective flight trajectory. a** The longitude-latitude plot of the case. A flying circle of the approach phase is caused due to traffic flow control near the arrival airport. A zoom-in local view is provided to show flight

trajectory predictions for the approach phase in complex airspace. **b** Visualization in the 3D grid illustrates that the selected trajectory contains climb, cruise, turn, descent and approach phases.

selected trajectory contains climb, cruise, turn, descent and approach phases. To obtain a more intuitive understanding of the differences between the models, the absolute error of the LLA and 3D deviation error are displayed in Fig. 4a–d. It is clear that the WTFTP framework achieves the desired prediction performance, whereas inferior results are obtained in baselines under time-varying flight patterns (various stages of the flight journey).

Specifically, A1–A3 can only reach the comparable performance during the cruise phase but fail to retain the prediction stability in the whole flight journey, subjecting to the intention changes as evident from Fig. 4d. In A4, the self-attention mechanism represents a lower prediction error and outperforms A1–A3. However, the performance is sharply dropped in the complex flight scenarios, particularly in the approach scene (flying circle), where traffic congestion and flow control result in the inability to accurately capture trajectory evolution patterns. Figure 3a provides the details of trajectory prediction in the approach area, indicating that the predicted trajectories of A1–A3 seriously deviate from the actual trajectory, and A4 also has a large prediction error. Meanwhile, when a left-turn intention is manifested in the historical trajectory, only the FlightBERT and WTFTP capture this motion detail and achieve a desired intention-driven response, providing a more accurate prediction. Although FlightBERT in A5 can perform comparable prediction accuracy, the high-bit prediction error also causes an unaccepted estimate error on the latitude dimension near the location of (102.06, 27.77). The high-bit errors of A5 are also indicated by red ellipses in Fig. 4a–c.

Thanks to wavelet-based time-frequency analysis for capturing flight trends and motion details, the WTFTP framework can capture in-depth flight patterns and achieve desirable FTP throughout the entire flight journey. Even in the approach area, the proposed WTFTP method is also able to harvest desired performance advantages compared to modern methods. During the approach phase, the baseline models suffer from large prediction errors due to the traffic control maneuver, especially for the MDE measurement. To quantitatively evaluate the performance in complex airspace (flying circle), the MDE and dynamic time warping (DTW)[55] metrics are calculated to clarify the performance improvements of the WTFTP in the approach phase:

(1) The MDE of the WTFTP framework is only 0.5003 km, while A1–A5 are specifically 3.6798, 2.1887, 2.1481, 1.2102 and 0.7873 km, respectively.

(2) As to the DTW metric, the proposed WTFTP framework is only 15.00 km, while A1–A5 are 94.60, 57.42, 56.91, 30.18 and 23.45 km, respectively.

Compared to the best baseline (FlightBERT), the WTFTP framework achieves over 36% relative reduction of both the MDE and DTW metrics. The results indicate the highest similarity between the prediction trajectory of the WTFTP framework and the ground truth, confirming that the proposed approach is able to provide excellent performance and is a promising FTP solution in complex airspace situations.

### Case study of multi-resolution features

To clarify the effects of multi-resolution features in the proposed WTFTP framework, a case study is conducted to investigate the learned flight trends and motion details by distinct WTCs for descending and turning right intentions. Figure 5j illustrates the flight profile, and $3 \times 3$ subgraphs in Fig. 5a–i show the estimated and ground-truth values of the LLA reconstructed using different WTCs. The subgraphs in the column represent the LLA, and each row indicates the involved WTC of the IDWT procedure (Not involved ones will be replaced by zeros) in the proposed WTFTP framework. For the 1-level wavelet analysis, the WTFTP yields two WTCs. There are thus a total of 3 cases for the IDWT: only $WTC_0$ involved (case 1), only $WTC_1$ involved (case 2), and all WTCs involved (case 3). Note that results in

case 3 can also be obtained by the sum of which in case 1 and 2. By analytical investigation and comparison of the mentioned three cases, the following conclusions can be made from the experimental results:

1. The WTCs are capable of implementing the time-frequency characterization. As can be seen from Fig. 5a–f, all position components of the trajectory present wave-like forms due to the absence of frequency information, which limits the ability to convert the trajectory fully into the time domain. In addition, the position components in case 2 exhibit larger fluctuations compared to those in case 1, resulting from that $WTC_0$ and $WTC_1$ capture low- and high-frequency features, respectively.

2. In general, $WTC_0$ primarily characterizes the global flight trends of the trajectory. With respect to the overall prediction results, the LLA in case 1 can roughly match the time-domain trajectory, indicating that $WTC_0$ retains the trend patterns of the trajectory sequence. Moreover, Fig. 5i depicts a zero rate of climb or descent (ROCD) of the aircraft between the time stamps 2 to 5, implying no local motions along the altitude during this period. Therefore, even without fast dynamic features (i.e., $WTC_1$), the altitude component in case 1 can closely reflect the time-domain altitude.

3. Compared to the $WTC_0$, the more local motion details of the trajectory sequence are represented by $WTC_1$. From the time stamp 0–7, the aircraft maintains its motion states along a straight flight without turning (i.e., heading changes), which results in fixed amplitudes in both the longitude and latitude components. At time stamp 8, the aircraft performs a right turn intention, causing prominent dynamics in the longitude dimension and slighter variance in the latitude dimension. Consequently, as shown in Fig. 5d, e, the corresponding changes in amplitude are concerned from time stamp 7–9. Compared to case 1, the altitude in case 2 is around zero between the time stamps 2–5, attributing to the zero ROCD and the absence of fast dynamics in this period.

By learning time-frequency features from both $WTC_0$ and $WTC_1$, the WTFTP framework achieves a comprehensive understanding of both global flight trends and local motion details in a given trajectory sequence, reaching the desired performance improvement in predicting the future trajectory. The IDWT procedure on all WTCs yields the time-domain trajectory in case 3. Even in such complex flight transitions, i.e., descending and turning, the WTFTP framework is still able to accurately reconstruct the raw flight trajectory, as well as predict the next trajectory position, which enhances the explainability of the proposed time-frequency analysis approach.

## Discussion

In this work, a time-frequency analysis framework is proposed to achieve flight trajectory prediction, providing a more dedicate perspective to promote the modeling capability of trajectory patterns. The proposed wavelet-transform based flight trajectory prediction (WTFTP) framework focuses on studying the virgin work of time-frequency analysis in the FTP research and addressing the disability of capturing both the global and local trajectory patterns in conventional methods. Firstly, inspired by frequency-domain analysis in other TSF tasks, the general time-frequency framework implemented by discrete wavelet transform is presented to optimize wavelet coefficients and support historical trajectory reconstruction and future state prediction. Secondly, the wavelet coefficients are generated by an encoder-decoder neural architecture from historical trajectory sequences, which are further fed into the IDWT procedure to achieve trajectory prediction. Finally, a wavelet attention module is introduced in the neural architecture to learn scale-oriented features and enhance the learning ability of the proposed model.

Experimental results have demonstrated that the WTFTP framework achieves a satisfactory performance improvement over selected competitive baselines on a real-world dataset. The results also indicate that each wavelet component contributes to the expected ability to

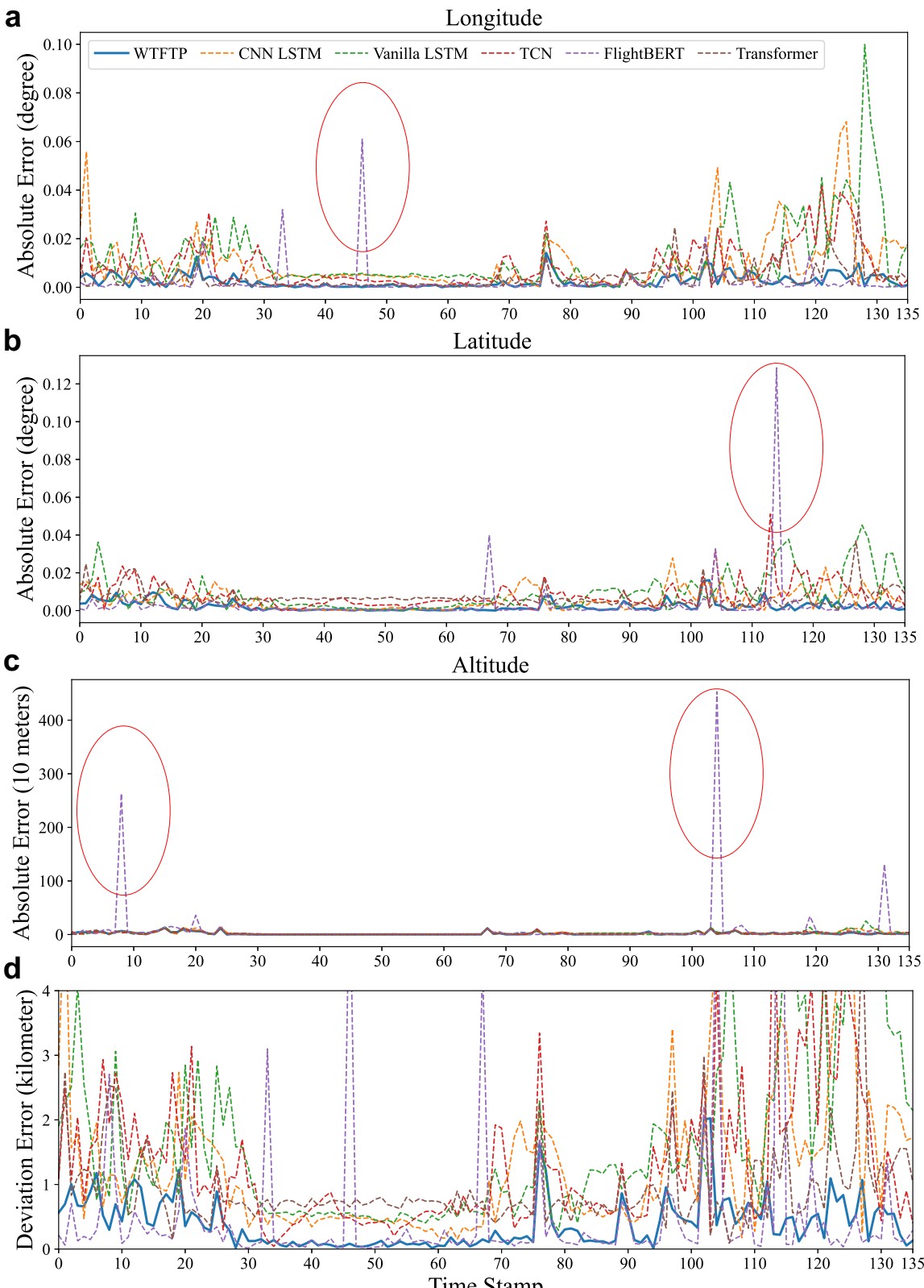

**Fig. 4 | Absolute error and deviation error of trajectory prediction with the WTFTP framework and baselines.** The red ellipses highlight abnormal predictions of the FlightBERT caused by high-bit errors. **a** The absolute error of longitude. **b** The absolute error of latitude. **c** The absolute error of altitude. **d** The 3D deviation error. The error data is provided as a source data file.

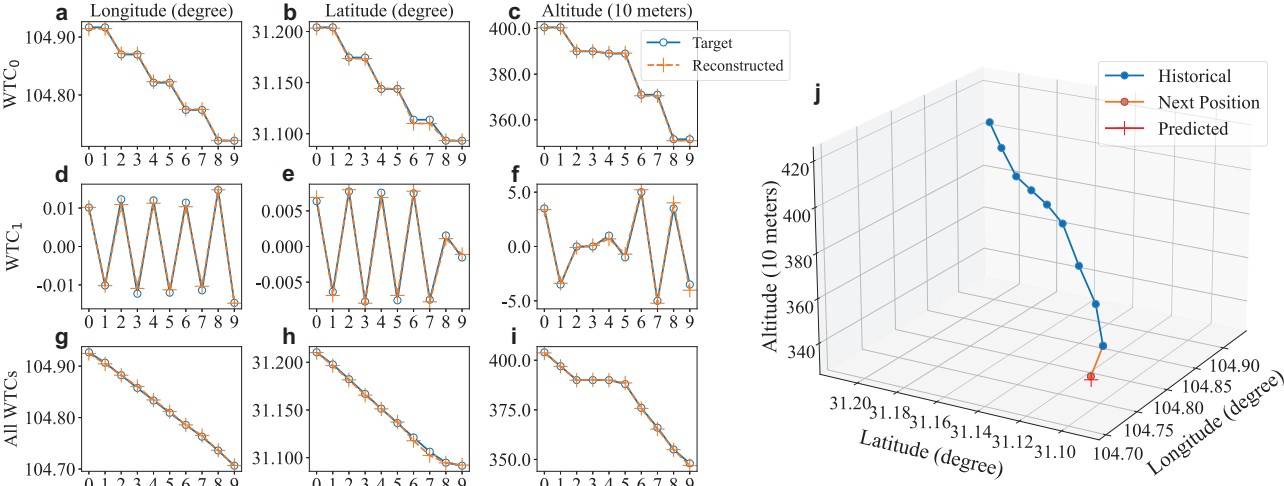

**Fig. 5 | The contributions of different WTCs for the FTP task. a–i** The target and predicted trajectory obtained by the IDWT procedure of different WTCs, in which *x*-axis represents time stamps and *y*-axis represents the position component. The subgraphs in the column represent the longitude, latitude and altitude. Each row indicates the involved WTC of the IDWT procedure (Not involved ones will be replaced by zeros) in the proposed WTFTP framework. **j** The selected trajectory in the 3D grid. All relevant coefficients and trajectory are provided as a source data file.

learn trajectory patterns at different scales, which confirms the effectiveness of time-frequency analysis in the FTP task. Furthermore, the WTFTP framework can achieve robust predictive stability for complex airspace situations, especially in the climb, descent and approach phases with maneuver control, which addresses the technical bottlenecks for conventional methods to retain high accuracy. Such performance improvements can be attributed that time-frequency analysis allows for an in-depth feature extraction toward global flight trends and local motion details. Meanwhile, the absence of time-frequency modeling poses a challenge for modern methods in promptly responding to maneuver control, which consequently limits the practicality in complex airspace.

Even though the WTFTP framework achieves significant performance improvement over comparative baselines, the following topics deserve to be further explored in our future works.

(1) It is required to enhance the prediction accuracy on the altitude dimension, especially in the cruise phase. As illustrated in Table 2, during the cruise phase, the improvement of the WTFTP framework in altitude is limited, and the three metrics are not comparable to other baseline models. Only during the climb and descent phases can the advantage of the WTFTP framework on the altitude dimension be achieved. As the major phase of the flight operation, the altitude dimension during the cruise phase is with limited maneuver control, the WTFTP framework may over-model the fast dynamics of the altitude changes, resulting in unnecessary estimation noise to degrade the prediction performance. In the future, we plan to control the convergence of different wavelet components and reduce the influence of high-frequency noise from the perspective of the loss function.

(2) The multi-step prediction of the proposed framework is a significant topic in future works. As shown in Supplementary Fig. 4, the mean deviation errors of the WTFTP framework and other baseline models at different prediction steps. Although the WTFTP framework maintains a higher performance within 80-second prediction horizons, it fails to outperform FlightBERT for longer prediction horizons. Given the ability of modeling local motion details by wavelet analysis, the WTFTP framework is sensitive to historical deviations in the iterative prediction procedure. The detailed multi-step prediction analysis is provided in Supplementary Section 3. In the future, we plan to incorporate the non-autoregressive mechanism into a multi-step prediction framework based on time-frequency analysis, which is expected to predict the aircraft state for future periods and avoid the accumulated impacts caused by pseudo labels.

Nevertheless, the proposed framework achieves higher performance over competitive baselines, which provides a time-frequency perspective to solve the FTP task by modeling local motion details and global flight trends. In addition, the proposed framework harvests pleasing results for maneuvering control, which addresses the technical bottlenecks of the time-domain methods.

## Methods
### Time-frequency features of flight trajectory
In this work, discrete wavelet transform is utilized to perform time-frequency analysis in the proposed framework. The global flight trends and local motion details of the flight trajectory can be accurately illustrated, benefiting from the filter bank obtained by wavelet transform. The preliminaries concerning wavelet analysis are provided in Supplementary Section 2.1. Specifically, the wavelet coefficients of each leaf node in the filter bank are defined as a wavelet component (WTC). These WTCs are sorted in ascending order of frequency. Without loss of generality, for the 3-level DWT, $WTC_0$ represents level-3 approximated coefficients and $\{WTC_{4-i}\}_{i \in (0,3] \cap \mathbb{Z}}$ represents level-*i* detail coefficients, as shown in Supplementary Fig. 2. By employing multi-resolution analysis of wavelet, $WTC_0$ is capable of illustrating the tendency of time series characterized by slow dynamics, while $\{WTC_{4-i}\}_{i \in (0,3] \cap \mathbb{Z}}$ present the local details of the series with fast dynamics[56]. Thanks to the ability of describing both the global and detailed dynamics by wavelet analysis, the wavelet-based time-frequency features of trajectories can be utilized to capture in-depth flight dynamic properties. Inspired by this, the WTFTP framework is proposed to learn global flight trends and local motion details from a time-frequency perspective, providing a more effective identification of flight patterns to further improve prediction accuracy.

### The proposed neural architecture
For the FTP task, the primary idea of the WTFTP framework is to predict the wavelet coefficients to implement the IDWT procedure, which reconstructs the input trajectory sequence and also predicts the trajectory point at the next time step. To this end, an encoder-decoder neural architecture is designed to implement the WTFTP framework,

as shown in Fig. 6, including an input embedding network, an encoder, multiple decoders corresponding to all WTCs, and an IDWT module. Unlike conventional FTP methods, the proposed WTFTP framework has the ability to predict the trajectory point at the next time step and also reconstruct the historical trajectory sequence, which enables it to fully learn multi-resolution characteristics of flight patterns and thus enhances the prediction accuracy.

In the proposed neural architecture, each historical trajectory point is firstly transformed into a high-dimensional vector via a fully connected network. The resulting high-dimensional vectors are fed into an RNN-based encoder to further model and extract the trajectory embeddings $\mathbf{H}$ as shown in Eq. (10). These embeddings capture the temporal features of the trajectory to support subsequent prediction steps for learning diverse flight patterns.

$$\mathbf{H} = \text{Encoder}(\text{InputEmbedding}(\mathbf{P}_{N-M:N-1}^{\mathsf{T}})) \quad (10)$$

In succession, multiple decoders are designed to achieve in-depth feature learning of flight patterns at different scales. For $L$-level wavelet analysis, each decoder is dedicated to generating a certain sub-band of time-frequency characteristics of the trajectory sequence, i.e., $\mathbf{Q}_i$ representing WTC$_i$ as shown in Eq. (11). To enhance the learning ability, a wavelet attention module is innovatively proposed in the decoder to combine the historical trajectory embeddings for generating scale-oriented features, which are then fed into an RNN-based block to learn the temporal dependence.

$$\mathbf{Q}_i = \text{Decoder}_i(\mathbf{H}), i = 0, 1, \cdots, L \quad (11)$$

Finally, the required attributes of the predicted trajectory point $\hat{\mathbf{P}}_N$ can be obtained by the IDWT procedure of stacked WTCs from multiple decoders as shown in Eq. (12).

$$[\hat{\mathbf{P}}_{N-M:N-1}, \hat{\mathbf{P}}_N] = \text{IDWT}(\mathbf{Q}_0, \mathbf{Q}_1, \cdots, \mathbf{Q}_L) \quad (12)$$

The MSE loss performed on the wavelet components is introduced to update the model parameters, which measures the difference between the predicted and actual wavelet component values to refine the MRA capability and reach the model convergence.

### Input embedding network

Each trajectory point represents a unique low-dimensional vector in the three-dimensional (3D) earth space via its positions and velocities. To fully extract implicit trajectory features for the subsequent networks, it is required to map the low-dimensional vector into a high-dimensional abstract feature space. Therefore, a linear mapping-based input embedding network is designed to boost the representational capability of the trajectory sequence, as shown below:

$$\mathbf{I} = \sigma(\sigma(\mathbf{P}_{N-M:N-1}^{\mathsf{T}} \cdot \mathbf{W}_{i1}) \cdot \mathbf{W}_{i2}) \quad (13)$$

where $\mathbf{I} \in \mathbb{R}^{M \times D}$ serves as high-dimensional abstract features of the input trajectory sequence $\mathbf{P}_{N-M:N-1} \in \mathbb{R}^{d \times M}$. $\mathbf{W}_{i1} \in \mathbb{R}^{d \times (D/2)}$ and $\mathbf{W}_{i2} \in \mathbb{R}^{(D/2) \times (D)}$ are weight matrices to linearly transform low-dimensional attribute vectors of trajectory points into high-dimensional abstract feature space. $\sigma(\cdot)$ is the ReLU activation function to enhance non-linear modeling ability. $M$ is the number of historical

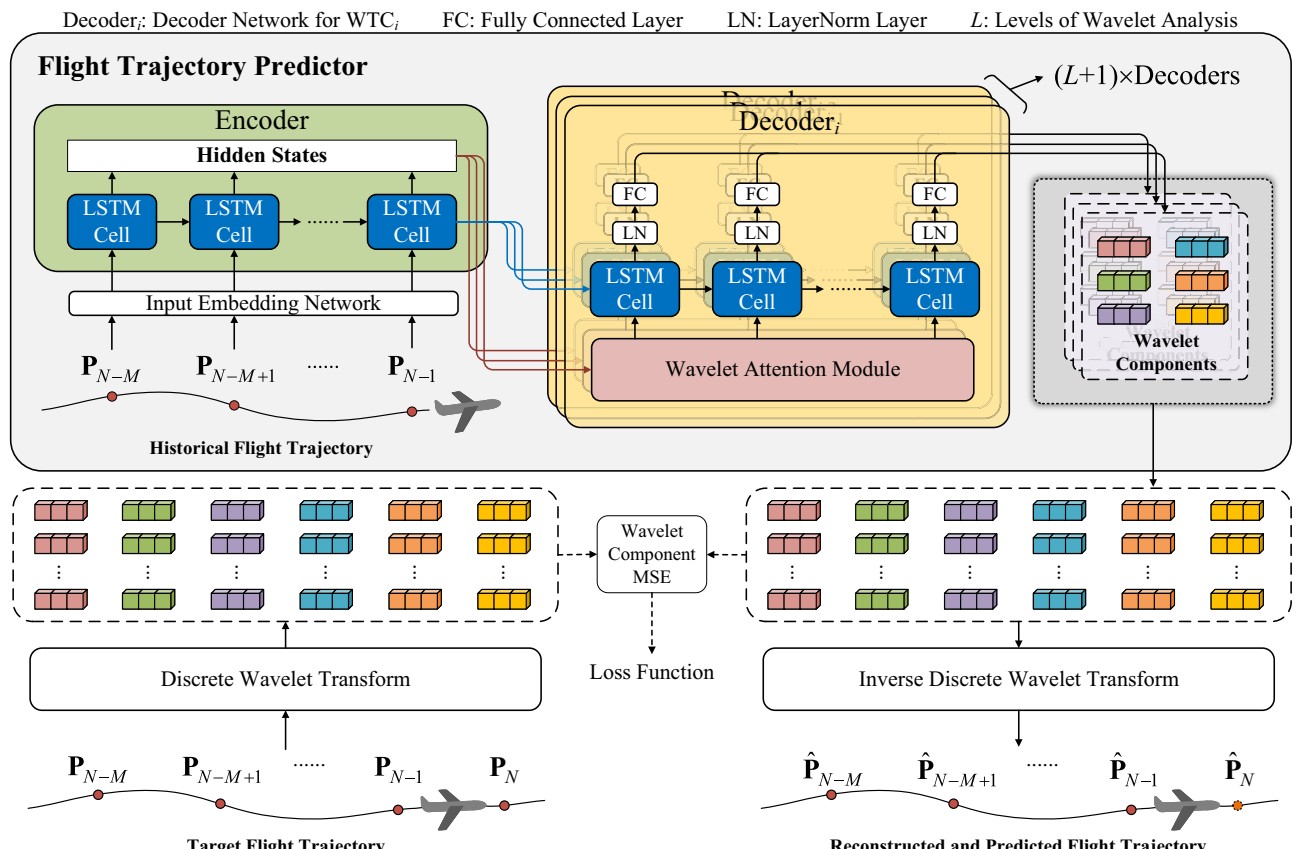

**Fig. 6 | The proposed neural architecture.** The network is cascaded by an input embedding network, an encoder, multiple decoders corresponding to all wavelet components, and an IDWT module. In each decoder, a wavelet attention module is devised to capture scale-oriented features of the trajectory sequence. The generated components by decoders illustrate global flight trends and local motion

details. An orthodox inverse discrete wavelet procedure further transforms wavelet components into flight trajectory sequence of the past period and next instant. The mean squared error of generated wavelet components serves as the loss function to update the trainable parameters of the neural networks.

trajectory points and $D$ denotes the dimension of feature space. Operator // is the floor division.

## Encoder

The input embedding network is able to capture the intuitive features of the trajectory points, such as the relationship among dynamic attributes of a single point. However, high-level abstract semantic features are highly required to achieve the FTP task within a certain trajectory sequence. To this end, an RNN-based encoder is utilized to build the temporal dependence to obtain trajectory embeddings with task-oriented dynamic characteristics, which enables a fine-grained analysis towards the intercorrelation of the trajectory sequence.

In this work, the LSTM block is selected to achieve the temporal modeling in the encoder. The generated high-dimensional features of the input embedding network are fed into the encoder, in which the produced hidden states will be considered as trajectory embeddings, as shown below:

$$\mathbf{H}, (\mathbf{h}_e, \mathbf{c}_e) = \mathrm{LSTM}(\mathbf{I}, \mathbf{h}_0, \mathbf{c}_0) \qquad (14)$$

where $\mathbf{h}_0, \mathbf{c}_0 \in \mathbb{R}^{S_1 \times D}$ are zero-initialized hidden state and cell state, respectively. $\mathbf{I} \in \mathbb{R}^{M \times D}$ is the input of the LSTM block. The output of the LSTM block consists of two components. $\mathbf{H} \in \mathbb{R}^{M \times D}$ saves all hidden states from the last layer of the LSTM block as the output features. The tuple of $(\mathbf{h}_e \in \mathbb{R}^{S_1 \times D}, \mathbf{c}_e \in \mathbb{R}^{S_1 \times D})$ denotes the hidden state representing short-term memory and the cell state representing long-term memory at the last time step, respectively. $S_1$ is the number of recurrent layers.

## Decoder

The encoder leverages the LSTM block to extract high-level abstract trajectory embeddings with rich temporal features, providing a robust characterization of the input trajectory sequence. For achieving the FTP in the proposed approach, the primary requirement is to predict the wavelet coefficients required for performing the IDWT procedure. It is believed that relying only on wavelet feature engineering, i.e., decomposing the sequence and feeding components into the prediction model, is hard to fully provide the potential of the multi-resolution representation offered by wavelet analysis like in refs. 57, 58.

As time-frequency representations of flight patterns, WTCs provide diverse multi-resolution dynamic features. In order to implicitly dissect trajectory embeddings and achieve an in-depth analysis of global trends and local details of flight patterns, a wavelet attention (WAtt) module is innovatively designed to learn scale-oriented features. The architecture of the WAtt module is illustrated in Fig. 7, which consists of two stages: enhancement process and convolution

operation. Specifically, trajectory embeddings obtained from the encoder are further weighted by attention scores to yield enhanced scale-oriented features of the corresponding time-frequency representation, i.e., enhanced trajectory embeddings. The WAtt module further performs convolution operations on the enhanced trajectory embeddings to generate contextual embeddings of WTCs. Mathematically, the enhancement process of the WAtt module can be represented as follows:

$$\mathbf{H}_p = \sigma(\mathbf{H} \cdot \mathbf{W}_{p1}) \cdot \mathbf{W}_{p2} \qquad (15)$$

$$\mathbf{E} = \gamma(\mathbf{W}_{s2} \cdot \sigma(\mathbf{W}_{s1} \cdot \mathbf{H}_p)) \qquad (16)$$

$$\mathbf{H}_h = \mathbf{H} + \mathrm{Diag}(\mathbf{E}) \cdot \mathbf{H} \qquad (17)$$

where $\mathbf{W}_{p1} \in \mathbb{R}^{D \times (D//2)}$ and $\mathbf{W}_{p2} \in \mathbb{R}^{(D//2) \times 1}$ are weight matrices to linearly transform trajectory embeddings into pooled features $\mathbf{H}_p \in \mathbb{R}^{M \times 1}$. $\mathbf{W}_{s1} \in \mathbb{R}^{(M//2) \times M}, \mathbf{W}_{s2} \in \mathbb{R}^{M \times (M//2)}$ are weight matrices to exploit feature importance of trajectory embeddings. $\sigma$ denotes the ReLU activation function, and $\gamma$ denotes the Sigmoid activation function. $\mathbf{E} \in \mathbb{R}^{M \times 1}$ activated by the Sigmoid function serves as attention scores to determine the importance of historical trajectory points with respect to the future trajectory point. The enhanced trajectory embeddings $\mathbf{H}_h \in \mathbb{R}^{M \times D}$ are inferred by combining the original trajectory embeddings and the weighted trajectory embeddings by $\mathbf{E} \in \mathbb{R}^{M \times 1}$. The operator $\mathrm{Diag}(\cdot)$ returns the matrix with the elements of input as the diagonal.

After obtaining the enhanced trajectory embeddings, the next step is to transform them into the contextual embeddings of the WTCs. To this end, the WAtt module utilizes one-dimensional convolution operations and expertly aligns the enhanced trajectory embeddings, which formulates highly correlated contextual embeddings $\mathbf{C} \in \mathbb{R}^{h \times D}$ with the same temporal dimension $h$ as the corresponding WTC. This step can be illustrated as follows mathematically:

$$\mathbf{C} = \sigma(\mathrm{Conv1d}(\mathbf{H}_h)) \qquad (18)$$

To determine the temporal dimension of $\mathbf{C}$, the length of $L$-level WTC obtained by the DWT is reduced to about $1/2^l$ of the original sequence length, due to the downsampling operations. In addition, the DWT in practice usually shifts the wavelet function to perform convolution operations on the time sequence. With the approaching of filters to the edges of a finite signal, the convolution operations require values beyond the signal boundaries through signal extension[59]. Therefore, the exact length of the WTC is determined by both the

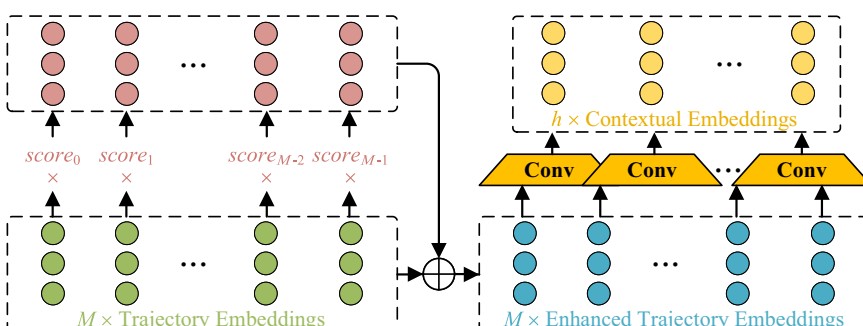

**Fig. 7 | Implementation details of wavelet attention module.** Conv denotes the convolution operation. The first stage is the enhancement process, in which the trajectory embeddings obtained from the encoder are weighted by attention scores to yield enhanced scale-oriented features, i.e., enhanced trajectory embeddings. The attention scores ($score_i, i = 0, 1, \cdots, M-1$) are calculated from the trajectory

embeddings to determine the importance of historical trajectory points. Further, the second stage is the convolution process, in which the enhanced trajectory embeddings are extracted into the contextual embeddings for generating wavelet components and aligned into the same temporal dimension as the corresponding wavelet components.

wavelet and signal extension mode. The symmetric extension, as a general extension mode[60], is selected to ensure continuity at the boundary of the signal. In this way, the length of the WTCs over $L$-level DWT in a convolution-based algorithm can be inferred through the following equations:

$$h_0 = \left\lfloor \frac{M}{2} \right\rfloor + l \tag{19}$$

$$h_i = \left\lfloor \frac{h_{i-1}-1}{2} \right\rfloor + l, i = 1, \cdots, L-1 \tag{20}$$

$$h_L = h_{L-1} \tag{21}$$

where $L$ is the level of wavelet analysis, and $l$ is half of the length of the selected wavelet filter in the DWT procedure. $M$ is the number of historical trajectory points. $\{h_i\}_{i \in [0,L) \cap i \in \mathbb{Z}}$ represents temporal length of high-frequency WTC$_{L-i}$. $h_L$ represents temporal length of low-frequency WTC$_0$. Operator $\lfloor \cdot \rfloor$ gives the largest integer less than or equal to the input.

After obtaining the contextual embeddings from the WAtt module, the LSTM block in the decoder further model the contextual embeddings to yield the wavelet embeddings, which are further mapped by a linear transformation to produce WTCs:

$$\mathbf{c}_i = [\mathbf{c}_e[-1, :], \mathbf{O}_1^\mathsf{T}]^\mathsf{T} \tag{22}$$

$$\mathbf{H}_w, (\mathbf{h}_d, \mathbf{c}_d) = \text{LSTM}(\mathbf{C}, \mathbf{O}_2, \mathbf{c}_i) \tag{23}$$

$$\mathbf{Q} = \text{FC}(\text{LN}(\mathbf{H}_w)) \tag{24}$$

where Eq. (22) employs last-layer cell states $\mathbf{c}_e[-1, :] \in \mathbb{R}^D$ passed from the encoder to initialize the first-layer cell states of the LSTM block with remaining cell states zeroed by $\mathbf{O}_1 \in \mathbb{R}^{(S_2-1) \times D}$, while the hidden state $\mathbf{O}_2 \in \mathbb{R}^{S_2 \times D}$ is zero-initialized. $S_2$ is the number of recurrent layers. The long-term memory from the encoder is retrieved here to initialize the LSTM block with a prior memory of the historical trajectory sequence, strengthening its ability to exploit scale-oriented features. The output features $\mathbf{H}_w \in \mathbb{R}^{h \times D}$ serve as the wavelet embeddings and pass through the LayerNorm layer $\text{LN}(\cdot)$ and the linear projection layer $\text{FC}(\cdot)$. The desired WTC, $\mathbf{Q} \in \mathbb{R}^{h \times d}$, are finally obtained.

### IDWT module and loss function

The IDWT module inversely transforms WTCs of each attribute and thus reconstructs the input historical trajectory sequence and also predict trajectory attributes of the next instant. The reconstruction filters are pre-defined and transform the coefficient matrix set $\{\mathbf{Q}_i \in \mathbb{R}^{h_{L-i} \times d}\}_{i \in [0,L] \cap i \in \mathbb{Z}}$ into the trajectory attributes series of the size $d \times (M+1)$. Specifically, in each attribute, the reconstruction filters iteratively combine pairs of both low- and high-frequency coefficients until a sequence of attributes in the time domain is recovered. Mathematically, for $j$-th attribute, the temporal sequence is obtained by:

$$\hat{\mathbf{P}}_{N-M:N}[j-1, :] = \text{Trim}(\text{IDWT}(\mathbf{Q}_0[:, j-1], \mathbf{Q}_1[:, j-1], \cdots, \mathbf{Q}_L[:, j-1])), j = 1, 2, \cdots, 6 \tag{25}$$

where dimensional indices of all matrices are starting from zero. $\text{Trim}(\cdot)$ is applied to crop out the redundant segments at the end of the time series due to the signal extension, i.e., only the first $M+1$ elements on the time dimension retained. Specifically, the first $M$ elements represent the reconstructed historical trajectory sequence and the

following element represents the predicted aircraft state at the next instant.

In order to facilitate different decoders to learn corresponding time-frequency representations of trajectory attributes at diverse scales, the wavelet loss function, i.e., the sum of the mean squared error of generated WTCs, is designed to update the network parameters as shown below:

$$\mathcal{L} = \sum_{k=0}^{L} \mathcal{L}_{wavelet}^k \tag{26}$$

$$\mathcal{L}_{wavelet}^k = \frac{1}{h_{L-k} \cdot d} \sum_{i=1}^{h_{L-k}} \sum_{j=1}^{d} \left( c_{i,j}^k - \hat{c}_{i,j}^k \right)^2 \tag{27}$$

where $L$ is the level of wavelet analysis, $h_{L-k}$ represents the length of the WTC$_k$ and $d = 6$ is the number of attributes in this work. In Eq. (27), $\hat{c}_{i,j}^k$ is the estimated value of the $i$-th element for WTC$_k$ of the $j$-th attribute output by the $(k+1)$-th decoder, while $c_{i,j}^k$ serves as the corresponding ground truth.

### Reporting summary

Further information on research design is available in the Nature Portfolio Reporting Summary linked to this article.

## Data availability

We are not authorized to publicly release the whole dataset used during the current study concerning safety-critical issues. Nonetheless, the processed example samples are available on https://zenodo.org/record/8238768 (ref. 61). Source data for all figures, except Fig. 3, are provided as Source data file. The trajectory sequence in Fig. 3 spans a wide range of locations, thus it cannot be made publicly available for the safety of China civil aviation. Source data are provided with this paper.

## Code availability

The PyTorch version of the WTFTP framework is publicly available on https://zenodo.org/record/8238768 (ref. 61).

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

## Acknowledgements

This work was supported by the National Natural Science Foundation of China (NSFC) under grants No. 62001315 (Y.L. received this fund), and U20A20161 (J.Z. and Y.L. received this fund), and by the Open Fund of Key Laboratory of Flight Techniques and Flight Safety, Civil Aviation Administration of China (CAAC) under Grant No. FZ2021KF04 (Y.L. and J.Z. received this fund), also by the Fundamental Research Funds for the Central Universities under Grant No. 2021SCU12050 (Z.Z., D.G. and Y.L. received this fund).

## Author contributions

Z.Z., D.G. and Y.L. conceived and led the research project. Z.Z. developed the framework. Z.Z, D.G. and Y.L devised neural architecture and wrote the paper. Z.Z. implemented the neural architecture and produced experimental results. Z.Z. and S.Z. conducted data preprocessing and collected the experimental results. All authors provided results discussions. Y.L. and J.Z. approved the submission and accepted responsibility for the overall integrity of the paper.

## Competing interests

The authors declare no competing interests.
