## [Peer Review File · Nature Communications]

Flight trajectory prediction enabled by time-frequency wavelet transformREVIEWER COMMENTS

Reviewer #1 (Remarks to the Author):

1. Please provide a detailed explanation of the data source and accuracy of the flight trajectory dataset. Usually, there are two data sources for longitude and latitude in the ATC system: Primary Survey Radar (PSR) and Automatic Dependent Survey-Broadcast (ADS-B). If Lon and Lat are measured in degree, the accuracy of PSR is around 0.002 degrees, and the accuracy of ADS-B is around 0.0001 degrees.

2. It is seen that Table 2 and Table 3 show the comparative results of the prediction performance on proposed method against others. But it is seen that WTFTP still has a relatively large MAE and RMSE of Lon and Lat. It is recommended to discuss why the predicted results of WTFTP are acceptable, as the accuracy of ADS-B can already reach 0.0001 degrees.

3. On page 13, "Given the attribute vectors of the past M trajectory points $\{T_i \in \mathfrak{R}^d | \dots\}$...". From formula " $T_{N-M:N-1} = \{T_{N-M}, T_{N-M+1}, \dots, T_{N-1}\}$ (8)", we can get $T_{N-M:N-1} \in \mathfrak{R}^{d \times M}$. But on page 15, " $T_{N-M:N-1} \in \mathfrak{R}^{M \times D}$ is the input of the LSTM block". $T_i \in \mathfrak{R}^D$ or $T_i \in \mathfrak{R}^d$? What is the correct dimension of $T_{N-M:N-1}$? The author should carefully check the inconsistency of mathematical notation in the article.

4. In formula (15), " $H, (h_e, c_e) =$ " should be " $H(h_e, c_e)$ ".

5. In the explanation of formula (15), " $T_{N-M:N-1} \in \mathfrak{R}^{M \times D}$ is the input of the LSTM block. $H \in \mathfrak{R}^{M \times D}$ contains all hidden states from the last layer of the LSTM block as the output features.", We can see that the input and output dimensions of the decoder are consistent. On page 15, "To fully extract implicit trajectory features for the subsequent networks, it is required to map the low-dimensional vector into a high-dimensional abstract feature space.", How to realize the mapping of feature from low-dimensional vector to high-dimensional space?

6. Authors should share source code of proposed model such as python etc., which contains README file and example dataset used in manuscript.

Reviewer #2 (Remarks to the Author):

Full Title: Flight Trajectory Prediction Using Wavelet Transform: A Time-Frequency Perspective

General Comment: This paper presents a methodology for predicting short-term aircraft flight paths using artificial intelligence-based methods. The prediction model has been developed using a neural encoder-decoder architecture. This study's originality lies mainly in using frequency-domain features, and more specifically, wavelet transforms, to predict flight trajectories. Overall, this is a good and well-written paper. I recommend it for publication, with a few minor corrections.

I may have a few questions/comments regarding the paper.

- Reading the article, the authors mention that their methodology works for short-term predictions (of the order of a few minutes). I wonder if it is possible to give a number. In addition, did an analysis was done in order to determine the maximum prediction time horizon?

- I think section 2.1 on the dataset deserves more detail. Where do the data come from? What type of aircraft are these data representative of? How many trajectories are used on average per day? Is using data for one day enough for the test set?

- In terms of methodology, how are atmospheric conditions (mainly temperature) and wind taken into account in prediction?

- If the methodology is only useful for short-term forecasts, how can it be useful from an air operator's point of view?

- Regarding the structure of the paper, I wonder why the authors decided to present the results before their proposed methodology. I was expecting a structure in which the methodology would be presented first, followed by the results. I find that talking about the results before the methodology makes reading the document less obvious.

- Furthermore, section 3 is entitled "Discussion" but seems to be more of a conclusion in summary form. I was expecting to read a real discussion from the authors, explaining the advantages and disadvantages of their methods compared to those in the literature.

Title: “Flight Trajectory Prediction Using Wavelet Transform: A Time-Frequency Perspective” (NCOMMS-23-17296)

Dear Reviewers,

Thanks for your efforts in reviewing our work entitled “Flight Trajectory Prediction Using Wavelet Transform: A Time-Frequency Perspective” (NCOMMS-23-17296) submitted to Nature Communications.

All your comments were highly insightful and enabled us to greatly improve the overall quality of this manuscript. In this version, we have carefully reviewed the comments and have revised the manuscript accordingly. The format of the manuscript is also adjusted based on the journal requirements.

In this submission, we are uploading our point-by-point response to the comments, a PDF document with track changes (the blue indicates added content, the red indicates deleted content), a clean main manuscript, another clean manuscript with Nature LaTeX template, Supplementary Information and the required checklists. It is noted that some contents in the original version are moved to “Supplementary Information” to meet the requirements of the manuscript format.

Hope this revision can address your concerns.

Best regards,

All the authors

Revision Summary

Dear Reviewers:

Thanks for your efforts in reviewing our work. All your comments were highly insightful and enabled us to greatly improve the overall quality of this manuscript. In this version, we have carefully reviewed the comments and have revised the manuscript accordingly, as summarized below:

1) The methodology section is re-written to address the review comment about the technical details, including the input embedding network, the temporal length of the DWT, the IDWT procedure and loss function, the equations, and figure captions, etc.

2) The discussion section is re-written to provide a comprehensive summary of the proposed framework, mainly concerning the analysis and limitations.

3) Throughout the paper, the main texts are revised to meet the requirements of the journal formatting, including abstract (about 150 words), figures and tables (a total of 10), section architecture, captions, etc.

4) The details of the dataset are provided in this revision, including the data type, data period, aircraft type, ranges of data items, etc.

5) Supplementary information is provided to further clarify the details of this work, including the data descriptions, the model configurations, the inference procedure of the proposed framework, the multi-step analysis, etc.

6) The source code and examples are prepared to improve the reproducibility of this work, and will be released after acceptance. Nevertheless, the README file is now available on GitHub <https://github.com/MusDev7/wtftp-model>.

7) The page index in the response letter follows the PDF document of the previous template.

Hope this revision can address all your concerns.

Reviewer 1#

1. Please provide a detailed explanation of the data source and accuracy of the flight trajectory dataset. Usually, there are two data sources for longitude and latitude in the ATC system: Primary Survey Radar (PSR) and Automatic Dependent Survey-Broadcast (ADS-B). If Lon and Lat are measured in degree, the accuracy of PSR is around 0.002 degrees, and the accuracy of ADS-B is around 0.0001 degrees.

Author Response: Thanks for your comments. It is important to improve the overall quality of this manuscript.

The flight trajectories are collected from the real ATC system in China by multi-source Secondary Surveillance Radar (SSR) and Automatic Dependent Surveillance-Broadcast (ADS-B), which are further fused to formulate the integrated trajectory dataset to conduct this research. As to accuracy of the dataset, the preprocessing procedures are conducted to minimize measurement errors. In general, the reliability and stability of the obtained integrated trajectory are better than those of the ADS-B data. The detailed explanation of the data source is provided in the main text, and other required details of dataset have been appended in **Supplementary Section 1.1**.

Revision: "In this work, the raw flight trajectories are collected by multi-source Secondary Surveillance Radar (SSR) and Automatic Dependent Surveillance-Broadcast (ADS-B) from a real-world ATC system in China. The flight trajectory dataset is formulated by fusing the multi-source flight trajectories to validate the proposed approach. In addition, the preprocessing steps are conducted to enhance the data quality, including data check, multi-source data parsing, track point and flight registering, multi-source data alignment, data filtering, and fusion." (Page 4)

2. It is seen that Table 2 and Table 3 show the comparative results of the prediction performance on proposed method against others. But it is seen that WTFTP still has a relatively large MAE and RMSE of Lon and Lat. It is recommended to discuss why the predicted results of WTFTP are acceptable, as the accuracy of ADS-B can already reach 0.0001 degrees.

Author Response: Thanks for your insightful comment. It is important to improve the overall quality of this manuscript.

In general, the primary goal of this work is to flight trajectory prediction (FTP), i.e., predict the short-term future flight trajectory (motion state of the aircraft) based on its existing trajectory sequence, rather than the aircraft tracking or situational awareness for the airspace. Technically speaking, they are two different tasks and the generated trajectories of the aircraft tracking serve as the raw data of the FTP task. Therefore, we believed that it is unfair to compare their performance directly, i.e., the prediction error of the FTP task v.s. the measurement error of the aircraft tracking. To clarify this point, the problem formulation is moved to **Section 2. Result (as Task Overview)** to ensure readers can properly understand the task. (Page 3 and 4)

As to the accuracy of the ADS-B, it is the real-time measurement error caused by the navigation system, signal attenuation, etc. (0.0001 degrees in about 1 second update interval). In this work, as stated in the manuscript, the prediction horizon is 20 seconds, and we achieved about 0.002-degree MAE prediction performance. In this work, we experimentally demonstrated that the proposed WFTFP framework can achieve better prediction results compared to other competitive baseline models in this dataset.

3. On page 13, "Given the attribute vectors of the past M trajectory points $\{T_i \in \mathbb{R}^d | \dots\}$ ". From formula " $T_{N-M:N-1} = \{T_{N-M}, T_{N-M+1}, \dots, T_{N-1}\}$ (8)", we can get $T_{N-M:N-1} \in \mathbb{R}^{d \times M}$. But on page 15, " $T_{N-M:N-1} \in \mathbb{R}^{M \times D}$ is the input of the LSTM block". $T_i \in \mathbb{R}^D$ or $T_i \in \mathbb{R}^d$? What is the correct dimension of $T_{N-M:N-1}$? The author should carefully check the inconsistency of mathematical notation in the article.

Author Response: Thanks for your insightful comments. It is important to improve the overall quality of this manuscript. We apologize for the inconsistency of mathematical notation in the methodology and any resulting misunderstandings.

Actually, the input dimension of the LSTM block is indeed $\mathbb{R}^{M \times D}$, but the notation should not be $T_{N-M:N-1}$. The input of the LSTM block is the features extracted from the historical sequence by the input embedding network. We have addressed the issues here and assigned new notations to identify the trajectory points and the input of the LSTM block of the encoder in the revised manuscript. In addition, we have re-checked the mathematical definitions, symbols,

formulas, etc. throughout the manuscript and corrected any errors where necessary. The resulting revisions can be found in the manuscript with track changes.

Revision: The past M trajectory points are defined as:

$$\{\mathbf{P}_i \in \mathbb{R}^d | i = N - M, N - M + 1, \dots, N - 1\}$$

Meanwhile, the historical trajectory sequence can be also shown as:

$$\mathbf{P}_{N-M:N-1} = [\mathbf{P}_{N-M}, \mathbf{P}_{N-M+1}, \dots, \mathbf{P}_{N-1}]$$

The input embedding network maps $\mathbf{P}_{N-M:N-1} \in \mathbb{R}^{d \times M}$ into high-dimensional abstract feature space:

$$\mathbf{I} = \sigma(\sigma(\mathbf{P}_{N-M:N-1}^T \cdot \mathbf{W}_{i1}) \cdot \mathbf{W}_{i2})$$

where $\mathbf{I} \in \mathbb{R}^{M \times D}$ serves as high-dimensional abstract features of the input trajectory sequence $\mathbf{P}_{N-M:N-1} \in \mathbb{R}^{d \times M}$. Notation explanation of \mathbf{W}_{i1} , \mathbf{W}_{i2} and σ can be found on the revised manuscript (Page 15). In the encoder, $\mathbf{I} \in \mathbb{R}^{M \times D}$ is also the input of the LSTM block:

$$\mathbf{H}, (\mathbf{h}_e, \mathbf{c}_e) = \text{LSTM}(\mathbf{I}, \mathbf{h}_0, \mathbf{c}_0)$$

4. In formula (15), " $H, (h_e, c_e) =$ " should be " $H(h_e, c_e)$ ".

Author Response: Thanks for your careful reading.

Actually, in the original version, the notation " H " is not a function. " $H, (h_e, c_e)$ " stand for two components of the LSTM output. The former is the hidden states from the last layer, i.e., H . The latter is the memory of the LSTM block, i.e., the tuple of (h_e, c_e) , which stores the short-term memory and long-term memory of the input trajectory sequence, respectively. In this revision, we revised the mathematical formatting to highlight the vectors or matrices. In addition, extra descriptions of the LSTM output are provided in this revision.

Revision:

$$\mathbf{H}, (\mathbf{h}_e, \mathbf{c}_e) = \text{LSTM}(\mathbf{I}, \mathbf{h}_0, \mathbf{c}_0)$$

The output of the LSTM block consists of two components:

1) $\mathbf{H} \in \mathbb{R}^{M \times D}$ saves all hidden states from the last layer of the LSTM block as the output features.

2) The tuple of $(\mathbf{h}_e \in \mathbb{R}^{S_1 \times D}, \mathbf{c}_e \in \mathbb{R}^{S_1 \times D})$ denotes the hidden state representing short-term memory and the cell state representing long-term memory at the last time step, respectively.
(Page 16)

5. In the explanation of formula (15), " $T_{N-M:N-1} \in \mathbb{R}^{M \times D}$ " is the input of the LSTM block. $H \in \mathbb{R}^{M \times D}$ contains all hidden states from the last layer of the LSTM block as the output features.", We can see that the input and output dimensions of the decoder are consistent. On page 15, "To fully extract implicit trajectory features for the subsequent networks, it is required to map the low-dimensional vector into a high-dimensional abstract feature space.", How to realize the mapping of feature from low-dimensional vector to high-dimensional space?

Author Response: Thanks for your insightful comments. It is important to improve the overall quality of this manuscript.

In this work, the mapping of feature from low-dimensional vector to high-dimensional space is implemented by a multi-layer perceptron with the ReLU activation. As you suggested, the mentioned descriptions are detailly provided in this revision.

Revision: "Therefore, a linear mapping-based input embedding network is designed to boost the representational capability of the trajectory sequence, as shown below:

$$\mathbf{I} = \sigma(\sigma(\mathbf{P}_{N-M:N-1}^T \cdot \mathbf{W}_{i1}) \cdot \mathbf{W}_{i2})$$

where $\mathbf{I} \in \mathbb{R}^{M \times D}$ serves as high-dimensional abstract features of the input trajectory sequence $\mathbf{P}_{N-M:N-1} \in \mathbb{R}^{d \times M}$. $\mathbf{W}_{i1} \in \mathbb{R}^{d \times (D//2)}$ and $\mathbf{W}_{i2} \in \mathbb{R}^{(D//2) \times D}$ are weight matrices to linearly transform low-dimensional attribute vectors of trajectory points into high-dimensional abstract feature space. $\sigma(\cdot)$ is the ReLU activation function to enhance non-linear modeling ability. M is the number of historical trajectory points and D denotes the dimension of feature space. Operator // is the floor division." (Page 15)

6. Authors should share source code of proposed model such as python etc., which contains README file and example dataset used in manuscript.

Author Response: Thanks for your comments. We will publicly release the source code of the proposed model and example samples on GitHub after acceptance. The **README** file is now accessible on <https://github.com/MusDev7/wtftp-model>.

Reviewer 2#

1. Reading the article, the authors mention that their methodology works for short-term predictions (of the order of a few minutes). I wonder if it is possible to give a number. In addition, did an analysis was done in order to determine the maximum prediction time horizon?

Author Response: Thank you for your insightful comments.

In fact, it is hard to provide a definite criterion for the maximum prediction horizon of short-term flight trajectory prediction (FTP) task, since the desired prediction horizon is dependent on the error tolerance, downstream tasks, and the exact applications. To quantify the prediction error (using the mean deviation error in the 3D earth space and mean absolute error of the altitude dimension) of the proposed model for the multi-step prediction, we consider the prediction performance by taking the predictions as input to achieve the multi-step prediction in an autoregressive manner. The experimental results are shown in Fig. 1 and Fig. 2.

Figure 1. Mean deviation errors of the WTFTP on 180-second prediction horizons.

Figure 2. Mean absolute errors of the altitude of the WTFTP on 180-second prediction horizons.

As can be seen from the results, the prediction errors of both measurements gradually increase with the increasing of the prediction horizon. To further consider the multi-step prediction error, the maximum prediction horizon is analyzed from the perspective of conflict detection with high-safety requirements, as shown below.

In general, a minimum distance between two aircraft is required for safety flight operation, i.e., 5 nautical miles (*nm*) in the horizontal plane and 1000 feet (*ft*) in the vertical dimension [1]. Consider that the proposed FTP model predicts the trajectories of two flights and the predicted points at the maximum prediction horizon exactly meet the minimum separations. As illustrated in Fig. 3, the green dash line and orange dash line are the predicted trajectory positions for aircraft A and B, respectively. To consider the prediction errors, a tolerance cylinder with centering aircraft based on the mean prediction error (MDE and MAE of altitude) can be obtained to illustrate the protection zone, denoting by their corresponding colors. To avoid potential conflict, the tolerance cylinders for the two aircraft should never overlap, i.e., at least 2.5 *nm* and 500 *ft* distance in the horizontal and vertical dimensions, respectively.

Figure 3. The predicted points (A and B) at the maximum prediction horizon. Cylinders with corresponding color cover the actual positions of aircraft.

However, to provide proper time to perform the conflict resolution, a separation distance in the horizontal plane is required to allow for maneuvers (ascent or descent) in a possible resolution advisory. For instance, in the case of flight level FL200-FL420, as instructed by the traffic collision avoidance system (TCAS), the aircraft should maintain a vertical distance of at least 600 *ft* after conflict resolution (also called vertical threshold for corrective resolution

advisory) [2]. Assuming that the horizontal velocity of the aircraft is 800 kilometers per hour (km/h) and the climb rate of the resolution process is 1500 feet per minute (ft/min), the reserved separation distance is given by:

$$\frac{600 \text{ ft}}{1500 \text{ ft/min}} \times 800 \text{ km/h} = 5.3333 \text{ km} = 2.8797 \text{ nm}$$

Therefore, the horizontal error of the FTP model at the maximum prediction horizon should be further less than $(5 - 2.8797)/2 = 1.0601 \text{ nm}$, i.e., 1.9633 kilometers (km). In summary, the preferred error tolerances for safety flight operation are 1.0601 nm (1.9633 km) and 500 ft (0.1524 km) in the horizontal and vertical dimensions, respectively, as illustrated in Fig. 4.

Figure 4. The desired tolerance zone allowing for conflict resolution.

Based on the aforementioned discussion, it is found that the proposed WTFTP framework is able to meet the requirements of the preferred vertical distance in 100 seconds. As to the horizontal plane at the 100-second prediction horizon, the prediction error is 1.9601 km ($\sqrt{1.9640^2 - 0.1235^2}$), which is also a safety separation (i.e., 1.9633 km). The horizontal errors are calculated by removing the height error from the MDE, and the earth curvature is not considered in the short term. Therefore, it is believed that the WTFTP framework has the ability to support the real-time conflict detection in 100-second prediction time horizon. Considering that the TCAS can approximately provide 48-second in advance [2], which allows the pilot to initiate a visual search and avoid potential intrusion. In this context, by incorporating proposed WTFTP framework into the TCAS, the pre-warning time is expected to be enhanced to support the ATC work, which further improves the airborne safety.

Reference:

[1] International Civil Aviation Organization. Separation Minima and Airspace Capacity (NO.WP05). Fourth Meeting of the South Asia/Indian Ocean ATM Coordination Group and the Twenty-First South East Asia ATM Coordination Group. Hong Kong, China, 2014-1-18. Retrieved from <https://www.icao.int/APAC/Meetings/2014%20SEACG21SAIOCG41/Forms/AllItems.aspx>.

[2] EUROCONTROL. ACAS Guide-Airborne Collision Avoidance Systems. 2022-3-25. Retrieved from <https://www.eurocontrol.int/publication/airborne-collision-avoidance-system-acas-guide>.

2. I think section 2.1 on the dataset deserves more detail. Where do the data come from? What type of aircraft are these data representative of? How many trajectories are used on average per day? Is using data for one day enough for the test set?

Author Response: Thanks for your insightful comment. It is important to improve the overall quality of this manuscript.

The raw trajectory data is collected from a real ATC system in China. The types of the aircraft cover Boeing 737-500, Boeing 737-800, Boeing 737-Max-8, Airbus A319, Airbus A320, etc., which are representative passenger aircraft of civil aviation. There are about 3200 trajectories used on average per day. Limited by the length of the main text, in this revision, the required details concerning our dataset are appended in the **Supplementary Section 1.1**. As to the test set, we apologize that the time range of the test set and the validation set are incorrectly provided in last version (actually, 1 day for validation and 4-day for test). We have addressed this issue in the revised manuscript.

Revision: "To evaluate the model performance, the trajectories in the first 40 days are selected as the training dataset, and the following one day is dedicated to fine-tuning hyper-parameters, and the trajectories of the last four days are formulated as a test set". (Page 4)

3. In terms of methodology, how are atmospheric conditions (mainly temperature) and wind taken into account in prediction?

Author Response: Thank you for your insightful comment. As illustrated in the original manuscript, flight trajectory prediction tasks can be categorized into short-term and long-term predictions in terms of prediction horizon. Short-term prediction mainly considers the motion state of the aircraft in past time, while long-term prediction is supported by external factors, such as flight intentions, wind, and atmospheric conditions in addition to the motion states.

Since this work focuses on the short-term prediction task, the future motion states are mainly inferred by modeling the historical trajectory sequence of the aircraft. Meantime, this work primarily investigates time-frequency analysis of the FTP task and dedicates to providing a new perspective to model the flight patterns of global trends and local details for the trajectory sequences. Therefore, the atmospheric conditions and wind have not been explicitly considered in this work.

As you suggested, the atmospheric conditions (mainly temperature) and wind are influential factors for the long-term prediction tasks. In future works, we will further investigate the feasibility of time-frequency analysis to consider the environmental factors in the proposed framework.

In this revision, the following descriptions are provided to clarify this point, as shown in **Section 1. Introduction.**

Revision: “In terms of prediction horizons, the FTP task can be classified into short-term and long-term trajectory prediction tasks [17]. Short-term prediction task aims to provide accurate positional estimation to infer immediate traffic situation, primarily by modeling historical flight trajectories to predict future motion states. As for the long-term prediction, additional external factors are required to support airspace operation planning and assessment, including flight intentions, meteorological conditions, wind speed, etc.” (Page 1 and 2)

4. If the methodology is only useful for short-term forecasts, how can it be useful from an air operator's point of view?

Author Response: Thanks for your insightful comment. Actually, the flight trajectory prediction (FTP) serves as a fundamental technique to improve the predictability of air traffic for the trajectory-based operation, which enables enhanced air-ground interconnection for a safe and

effective air traffic control (ATC). The downstream task of the FTP work concerns the estimation of arrival time, conflict detection and air traffic flow prediction, which are able to greatly reduce the workload of ATC controllers. In term of short-term FTP task, a crucial task is to achieve the real-time conflict detection. As discussed in **Comment 1#**, it is believed that the WTFTP can meet the safety requirements of the real-time conflict detection in a 100-second look-ahead horizon. Compared to the traffic collision avoidance system (TCAS), the WTFTP is able to provide more pre-warning for the aircrew to cope with any emergency situations, which can be applied to enhance the TCAS system.

Based on the high-confidence trajectory prediction, the ATC controllers are allowed to detect any potential conflicts in advance, improving overall operational safety and efficiency of airspace system. Therefore, by combining the proposed trajectory prediction model, the reliability, robustness, and accuracy of the downstream tasks can be greatly enhanced to support the practical ATC work. Most importantly, the proposed WTFTP framework contributes desired performance for maneuver control, which further supports the decision-making system to timely update maneuver tactics for guaranteeing the safety of local airspace.

5. Regarding the structure of the paper, I wonder why the authors decided to present the results before their proposed methodology. I was expecting a structure in which the methodology would be presented first, followed by the results. I find that talking about the results before the methodology makes reading the document less obvious.

Author Response: Thank you for your comments. Based on the formatting standards of the journal and the editor's suggestions, the manuscript is required to be organized in this way (**Section 2. Results** before **Section 4. Methodology**).

To clarify the task nature, in this revision, we have reorganized the sections of the manuscript. **Section 2.1. Task Overview** has been placed in **Section 2. Results** to describe the optimization objective of flight trajectory prediction and the task of the WTFTP framework, consequently facilitating the understanding of the experimental results.

6. Furthermore, section 3 is entitled “Discussion” but seems to be more of a conclusion in summary form. I was expecting to read a real discussion from the authors, explaining the advantages and disadvantages of their methods compared to those in the literature.

Author Response: Thanks for your insightful comments. It is important to improve the overall quality of this manuscript.

Based on the formatting requirements of this journal, the discussion section is re-written to summarize the key findings of the study and their significance in the context of flight trajectory prediction. Specifically, a dedicated section is also provided to discuss the limitations of this work and possible solutions for future studies. In general, the proposed framework is based on the time-frequency analysis to model the global trend and local details of the flight trajectory sequence. The superiority of the method and the effectiveness of the time-frequency analysis are confirmed by extensive experimental results. Even though the proposed framework harvests better performance over comparative baselines, it still suffers from the following limitations:

1) As can be seen from the results, the WTFTP model only achieves marginal performance improvement on the prediction of the altitude component.

2) In addition, to investigate the prediction horizon, we also consider the multi-step FTP task in future 3-minute (9 steps). The experimental results demonstrated that the improvement of the WTFTP framework is limited on multi-step prediction task.

As provided in the main text, in future works, we will attempt to address the mentioned limitations to further enhance the applicability of the proposed approach. In this revision, the mentioned analysis has been presented in the **Section 3. Discuss** of the main text, in which some of them are provided in **Supplementary Section 3** due to the space limitation.

Revision: “In this work, a time-frequency analysis framework is proposed to achieve flight trajectory prediction, providing a new perspective to promote the modeling capability of trajectory patterns. The proposed wavelet-transform based flight trajectory prediction (WTFTP) framework focuses on studying the virgin work of time-frequency analysis in the FTP research and addressing the disability of capturing both the global and local trajectory patterns in conventional methods. Firstly, inspired by frequency-domain analysis in other TSF tasks, the

general time-frequency framework implemented by discrete wavelet transform is presented to optimize wavelet coefficients and support historical trajectory reconstruction and future state prediction. Secondly, the wavelet coefficients are generated by an encoder-decoder neural architecture from historical trajectory sequences, which are further fed into the IDWT procedure to achieve trajectory prediction. Finally, a wavelet attention module is introduced in the neural architecture to learn scale-oriented features and enhance the learning ability of the proposed model.

Experimental results have demonstrated that the WTFTP framework achieves a satisfactory performance improvement over selected competitive baselines on a real-world dataset. The results also indicate that each wavelet component contributes to the expected ability to learn trajectory patterns at different scales, which confirms the effectiveness of time-frequency analysis in the FTP task. Furthermore, the WTFTP framework can achieve robust predictive stability for complex airspace situations, especially in the climb, descent and approach phases with maneuver control, which addresses the technical bottlenecks for conventional methods to retain high accuracy. Such performance improvements can be attributed that time-frequency analysis allows for an in-depth feature extraction toward global flight trends and local motion details. Meanwhile, the absence of time-frequency modeling poses a challenge for modern methods in promptly responding to maneuver control, which consequently limits the practicality in complex airspace.

Even though the WTFTP framework achieves better performance over comparative baselines, the following topics deserve to be further explored in our future works.

- 1) It is required to enhance the prediction accuracy on the altitude dimension, especially in the cruise phase. As illustrated in Table 2, during the cruise phase, the improvement of the WTFTP framework in altitude is limited, and the three metrics are not comparable to other baseline models. Only during the climb and descent phases can the advantage of the WTFTP framework on the altitude dimension be achieved. As the major phase of the flight operation, the altitude dimension during the cruise phase is with limited maneuver control, the WTFTP framework may over-model the fast dynamics of the altitude changes, resulting in unnecessary estimation noise to degrade the prediction performance. In the future, we plan to control the

convergence of different wavelet components and reduce the influence of high-frequency noise from the perspective of the loss function.

2) The multi-step prediction of the proposed framework is a significant topic in future works. As shown in Supplementary Fig. 4, the mean deviation errors of the WTFTP framework and other baseline models at different prediction steps. Although the WTFTP framework maintains a higher performance within 80-second prediction horizons, it fails to outperform FlightBERT for longer prediction horizons. Given the ability of modeling local motion details by wavelet analysis, the WTFTP framework is sensitive to historical deviations in the iterative prediction procedure. The detailed multi-step prediction analysis is provided in Supplementary Section 3. In the future, we plan to incorporate the non-autoregressive mechanism into a multi-step prediction framework based on time-frequency analysis, which is expected to predict the aircraft state for future periods and avoid the accumulated impacts caused by pseudo labels.

Nevertheless, the proposed framework achieves higher performance over competitive baselines, which provides a new perspective to solve the FTP task by modeling local motion details and global flight trends. In addition, the proposed framework harvests pleasing results for maneuvering control, which addresses the technical bottlenecks of the time-domain methods.” (Page 13 and 14)

REVIEWERS' COMMENTS

Reviewer #1 (Remarks to the Author):

The authors have revised the paper according to the comments given and all of my queries are well answered. I really appreciate the efforts made to improve the quality of the manuscript and I'm satisfied with it.

Reviewer #2 (Remarks to the Author):

The authors have made the suggested corrections and improved the quality of the paper, taking my comments into account. I believe this new version of the paper can be accepted for publication.

Reviewer 1#

1. The authors have revised the paper according to the comments given and all of my queries are well answered. I really appreciate the efforts made to improve the quality of the manuscript and I'm satisfied with it.

Author Response: We greatly appreciate the reviewers' time. We are pleased that our revision has addressed all your concern.

Reviewer 2#

1. The authors have made the suggested corrections and improved the quality of the paper, taking my comments into account. I believe this new version of the paper can be accepted for publication.

Author Response: We greatly appreciate the reviewers' time and all the helpful remarks during the revision.